# DNA double-strand break-free CRISPR interference delays Huntington's disease progression in mice

Jung Hwa Seo [1,2], Jeong Hong Shin[2,3], Junwon Lee[4], Daesik Kim[5], Hye-Yeon Hwang[5], Bae-Geun Nam[1,6], Jinu Lee [7], Hyongbum Henry Kim [2,3,6,8] & Sung-Rae Cho [1,2,6,9 ✉]

Huntington's disease (HD) is caused by a CAG repeat expansion in the huntingtin (HTT) gene. CRISPR-Cas9 nuclease causes double-strand breaks (DSBs) in the targeted DNA that induces toxicity, whereas CRISPR interference (CRISPRi) using dead Cas9 (dCas9) suppresses the target gene expression without DSBs. Delivery of dCas9-sgRNA targeting CAG repeat region does not damage the targeted DNA in HEK293T cells containing CAG repeats. When this study investigates whether CRISPRi can suppress mutant HTT (mHTT), CRISPRi results in reduced expression of mHTT with relative preservation of the wild-type HTT in human HD fibroblasts. Although both dCas9 and Cas9 treatments reduce mHTT by sgRNA targeting the CAG repeat region, CRISPRi delays behavioral deterioration and protects striatal neurons against cell death in HD mice. Collectively, CRISPRi can delay disease progression by suppressing mHtt, suggesting DNA DSB-free CRISPRi is a potential therapy for HD that can compensate for the shortcoming of CRISPR-Cas9 nuclease.

[1] Department and Research Institute of Rehabilitation Medicine, Yonsei University College of Medicine, Seoul, Republic of Korea. [2] Brain Korea 21 PLUS Project for Medical Science, Yonsei University College of Medicine, Seoul, Republic of Korea. [3] Department of Pharmacology, Yonsei University College of Medicine, Seoul, Republic of Korea. [4] Department of Ophthalmology, Institute of Vision Research, Yonsei University College of Medicine, Seoul, Republic of Korea. [5] Department of Precision Medicine, Sungkyunkwan University School of Medicine, Suwon, Republic of Korea. [6] Graduate Program of Biomedical Engineering, Yonsei University College of Medicine, Seoul, Republic of Korea. [7] College of Pharmacy, Yonsei Institute of Pharmaceutical Sciences, Yonsei University, Incheon, Republic of Korea. [8] Severance Biomedical Science Institute, Yonsei University College of Medicine, Seoul, Republic of Korea. [9] Rehabilitation Institute of Neuromuscular Disease, Yonsei University College of Medicine, Seoul, Korea. ✉email: srcho918@yuhs.ac

Huntington's disease (HD) is a progressive neurodegenerative disease caused by a CAG repeat expansion at the exon 1 location of the huntingtin (HTT) gene[1,2]. HD accompanied by impairments of motor function, cognition, and psychiatric symptoms can often lead to death[3,4]. The hallmark pathological feature of HD is the loss of striatal GABAergic medium spiny neurons caused by the presence of mutant huntingtin (mHTT) proteins. The mHTT protein leads to the imbalance of apoptotic homeostasis, dysfunction of mitochondria, increase of glutamate excitotoxicity, impairment of axonal transport, and abnormal transcription of brain derived from neurotrophic factor (BDNF)[5]. Such prior findings suggest that the wtHTT protein is important in protection against neurodegeneration.

In the recent search for a cure for HD, several approaches for selectively inhibiting mutant allele expression have been studied, including gene-silencing strategies such as RNA interference (RNAi), antisense oligonucleotides (ASOs), and genome editing tools. Allele-specific suppression methods such as RNAi to selectively inhibit mutant alleles without silencing the wtHTT gene have been attempted[6–8]. ASOs are undergoing clinical trials based on positive results derived in mouse and primate models[9,10]. However, in the clinical trials of ASOs in HD, Roche's phase III drug Tominersen reduced HTT levels but was halted due to a poor risk-benefit profile, while Wave's drugs (WVE-12010 and WVE-120102) did not induce effective silencing of mHTT mRNA[11]. Unlike RNAi, genome editing system directly targeting the DNA can lead to the permanent repair of specific genes with the potential for greater efficacy[9].

Since wtHTT is known to play a crucial role in the brain[5,12–15], allele-selective zinc finger protein transcription factors to target mutant CAG repeat region for the treatment of HD showed transcription repression of mHTT and neuropathological improvement in vitro and in vivo[16]. In addition, the CRISPR-Cas9 system has been used to suppress endogenous mHTT in the striatum of a mouse model of HD delaying neurological declines[17,18]. Although a previous study demonstrated that mHTT can be specifically inactivated by the use of a personalized allele-specific CRISPR-Cas9, this method is only applicable to patients with single nucleotide polymorphisms that affect protospacer adjacent motif (PAM) sites[19].

The latest studies have shown that DNA double-strand breaks (DSBs) induced by CRISPR-Cas9 lead to p53-mediated cell cycle arrest and cell death[20–22]. Moreover, Cas9-induced DNA DSBs can cause mutagenesis by indels at the target site and unintentional off-target sites. On the other hand, CRISPR interference (CRISPRi) system based on catalytically inactive dead Cas9 (dCas9), which does not induce DNA DSBs, allows specific repression of gene expression by sgRNA binding to target genomic loci. It represents an alternative approach to address potential challenges in genetic disorders including inherited retinitis pigmentosa[23,24]. Therefore, we considered a strategy to inhibit mHTT expression using CRISPRi that does not cause DSBs.

CRISPR systems require single guide RNA (sgRNA) to bind target DNA. The target sites should include a PAM sequence, whereby the most commonly used PAM is NGG. Despite the lower efficiency compared to NGG, various PAMs including NAG, NGA, and NCC have been experimented[25,26]. Therefore, we used sgRNA containing CAG PAM sequence to target CAG repeat region in treating HD.

Since wtHTT is thought to play an important role in the brain as a therapeutic strategy, we designed a dCas9 and sgRNA complex that binds to the CAG repeat region in HTT gene and represses mHTT transcription of the mutant allele relative to that of the wild-type allele. CRISPR/Cas9 system recognizes 23 bp including the PAM sequence. Theoretically, there should be at least 8 CAG repeats for Cas9 nucleases to bind to (3 bp repeat × 8 = 24 bp). Therefore, our mHTT suppression strategy was designed with the expectation that the longer the CAG repeat, the greater the suppression.

The wild-type CRISRP-Cas9 can be sufficient for genome editing in a short duration but CRISPRi strategy using dCas9 requires the sustained expression of dCas9 to inhibit transcription. Therefore, we used a lentiviral vector to constitutively express dCas9 in the HD models because the effect of dCas9 expression is not permanent.

This study investigated the therapeutic effects of dCas9 with sgRNA targeting the CAG repeat region in human HD fibroblasts and HD transgenic mice. We asked if delivery of dCas9-sgRNA suppresses mHTT in human HD fibroblasts, and the dCas9 treatment can mitigate behavioral deterioration in a mouse model of HD. Here, it has been shown that DNA DSB-free CRISPRi is a potential therapy for HD that can compensate for the shortcoming of CRISPR-Cas9 nuclease in reducing cell death and delaying disease progression.

## Results

### Delivery of dCas9 with sgRNA targeting the CAG repeat region does not damage the targeted DNA.
Our approach is based on the idea that increased binding of a CRISPRi system to mHTT versus wtHTT (as a result of the greater number of CAG repeats in the mutant allele) would repress mHTT expression relative to that of wtHTT (Fig. 1a). For expression of CRISPRi components, lentiviral vectors containing sgRNA that targets the CAG repeat region and either Cas9 or dCas9 derived from Streptococcus pyogenes (SpCas9) were used (Fig. 1b). Hereafter, this study refers to the lenti-SpCas9-sgRNA and lenti-SpdCas9-sgRNA constructs as Cas9-sgRNA and dCas9-sgRNA, respectively. To test the efficacy of the CAG repeat-targeting sgRNA, Cas9-sgRNA (with CAG PAM) and a reporter plasmid containing the sgRNA target site were co-transfected into HEK293T cells. If Cas9-sgRNA make DSBs in the target site in the reporter, nucleotide insertions or deletions at the target site can lead to eGFP expression. Our results show that 34% eGFP expressions were observed in HEK293T cells treated with Cas9-sgRNA. These results suggest that Cas9-RNA with CAG repeat-targeted sgRNA significantly acts on target site to make DSBs (Fig. 1c). To determine if the number of CAG repeats is specifically reduced by treatment with Cas9-sgRNA, Cas9-sgRNA or dCas9-sgRNA, we co-transfected with the plasmid containing target sites complementary to the sgRNA into HEK293T cells. The target CAG repeat region was deep sequenced at 3 days after transfection and it was observed that treatment with Cas9-sgRNA, but not dCas9-sgRNA, reduced the number of repeats through cleavage from 27 to 3–11 (Fig. 1d, e and Supplementary Fig. 1a–e). Therefore, dCas9 with sgRNA targeting the CAG repeat region circumvents the risk of altering the DNA sequence, a side effect of administering Cas9 nuclease as complicated by the fact that other CAG repeats in the genome would also be targeted.

### CRISPRi targeting the CAG repeat region decreases mHTT expression while preserving wtHTT expression.
To investigate the effects of this CRISPRi system in HD cells, we used six human fibroblasts: normal fibroblast (GM04775, 24 and 17 CAG repeats in normal allele; GM07492, 21 and 18 CAG repeats in normal allele), HD patient fibroblasts containing 40–50 CAG repeats (GM04022 in which the mHTT allele contains 44 CAG repeats versus the 18 repeats in the normal allele; GM04855 in which the mHTT allele contains 48 CAG repeats versus the 20 repeats in the normal allele), GM09197 in which the mHTT allele contains 180 CAG repeats (versus the 21 repeats in the normal allele), and

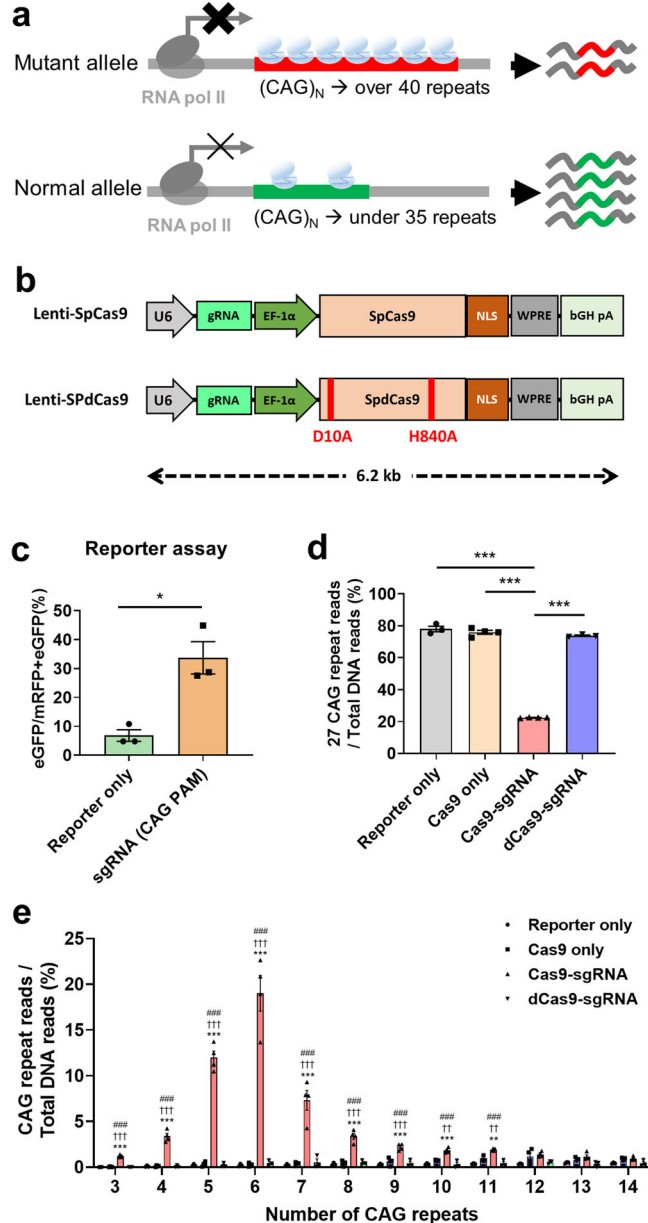

**Fig. 1 The sgRNA with CAG as PAM sequence reduces the number of CAG repeats. a** Schematic illustrating the expected phenotypes according to the relative repression of HTT. A CRISPRi was designed to bind to the polyglutamine tract and aims to reduce mHTT expression without altering CAG repeat length. Relative repression between the mutant allele and normal allele is caused by the different number of CAG repeats. Cas9 complexes bind to the mutant allele more than the normal allele due to the elongated polyglutamine stretch. **b** Schematic representation of the lentiviral vector containing indicated components. U6: human U6 polymerase III promoter; EF1a: elongation factor 1a promoter; NLS: nuclear localization signal; WPRE: posttranscriptional regulatory element; bGHpA: bovine growth hormone polyadenylation signal. **c** Flow cytometry of HEK293T cells 3 days after co-transfection of a plasmid encoding both the SpCas9 with sgRNA(CAG PAM) targeting CAG repeats and the surrogate reporter, or surrogate reporter only. Frameshift generated by indels can read to eGFP expressions changing out-of-frame to in-frame codons from mRFP expressions. The percentage of eGFP⁺/mRFP⁺eGFP⁺ cells is indicated that Cas9 with CAG-targeting sgRNA (CAG PAM) is highly efficient compared with Reporter only-treated cells ($n = 3$, each). Error bars represent the mean ± S.E.M. **$P < 0.01$, by independent t-test. **d** In HEK293T cells 3 days after co-transfection of plasmid including Reporter only ($n = 3$), Cas9 only ($n = 4$) with reporter, Cas9-sgRNA ($n = 4$) with reporter or dCas9-sgRNA ($n = 3$) with reporter. we deep sequenced the target CAG repeat region and observed that treatment with Cas9-sgRNA, but not the other groups including dCas9-sgRNA-treated cells, significantly reduced the number of repeats. Normalized ratio of DNA sequencing reads with 27 repeats of CAG. **e** Distribution of reduced CAG repeats caused by each experiment group. The x-axis indicates the number of CAG repeats. Error bars represent the mean ± S.E.M. **$P < 0.01$ and ***$P < 0.001$ versus Rep.+dCas9-sgRNA; ††$P < 0.01$ and †††$P < 0.001$ versus Rep.+Cas9 only; ##$P < 0.01$, ###$P < 0.001$ versus Reporter only, by one-way ANOVA with Bonferroni comparison.

GM21756 in which the *mHTT* allele contains 70 CAG repeats (versus the 15 repeats in the normal allele).

We first searched the potential off-target sites via Cas9-OFFinder algorithm. Cas-OFFinder algorithm was then used to analyze potential off-target sites for sgRNA targeting CAG repeat region in HTT gene. We obtained 211 sites for sgRNA targeting CAG repeat region from Cas-OFFinder by allowing for up to 3 bp mismatches and no DNA or RNA bulges (Supplementary data 1). When we identified mRNA expression of sorted genes, off-target effects in Abhydrolase domain containing 1 (*ABHD1*), Atrophin 1 (*ATN1*), Ataxin 3 (*ATXN3*), Decapping mRNA 1B (*DCP1B*), Forkhead box J2 (*FOXJ2*), Frizzled class receptor 1 (*FZD1*), Lysine methyltransferase 2D (*KMT2D*), Plectin (*PLEC*), SATB homeobox 1 (*SATB1*), Zinc finger protein 384 (*ZNF384*), Zinc finger protein 395 (*ZNF395*), and Zinc finger protein 853 (*ZNF853*) were not observed in GM09197 fibroblasts treated with dCas9-sgRNA. The on-target gene, HTT was also significantly decreased in dCas9-sgRNA-treated cells compared to Cas9-only-treated cells (Fig. 2a).

In addition, we confirmed several genes containing CAG repeats associated with polyglutamine-associated diseases[27,28], such as Ataxin1 *(ATXN1)*, *ATN1*, Ataxin1 *(ATXN7)*, calcium voltage-gated channel subunit alpha1 A *(CACNA1A)*, TATA-box binding protein *(TBP)*, and on-target gene, *HTT* in both human HD and human control fibroblasts. In human control fibroblasts, after Cas9-sgRNA treatment, the expression of *ATXN1*, *ATN1*, and *HTT* genes significantly decreased compared to the Cas9 only- and dCas9-sgRNA treatment. In HD fibroblast containing 40–50 CAG, *ATN1*, and *HTT* were significantly suppressed in the Cas9-sgRNA-treated group compared to the other groups and *HTT* gene expression was decreased in the dCas9-sgRNA group than Cas9 only group. The Cas9-sgRNA and dCas9-sgRNA significantly reduced the HTT gene in GM21756 cell line (HD fibroblasts) compared to Cas9 only group, and treatment with Cas9-sgRNA suppressed *HTT* gene expression more than treatment with dCas9-sgRNA. In GM09197 fibroblast, expression of *ATN1* and *TBP* genes was significantly suppressed by the Cas9-sgRNA treatment and *HTT* gene expression was significantly decreased in both the Cas9-sgRNA group and dCas9-sgRNA group compared to Cas9 only group. These results showed that suppression of the potential off-target genes was not observed in the dCas9-sgRNA-treated group and suggested that CRISPR-Cas9 cuts the DNA to shut the targeted gene off, but sgRNA targeting CAG repeat regions may have side effects of mutating normal genes. Consequently, both in silico and qRT-PCR for our sgRNA were used to identify potential off-targets (Supplementary Fig. 2a–d).

Digenome-seq is one method for profiling genome-wide off-target sites of Cas9 nuclease[29]. Using Digenome-seq, we were able

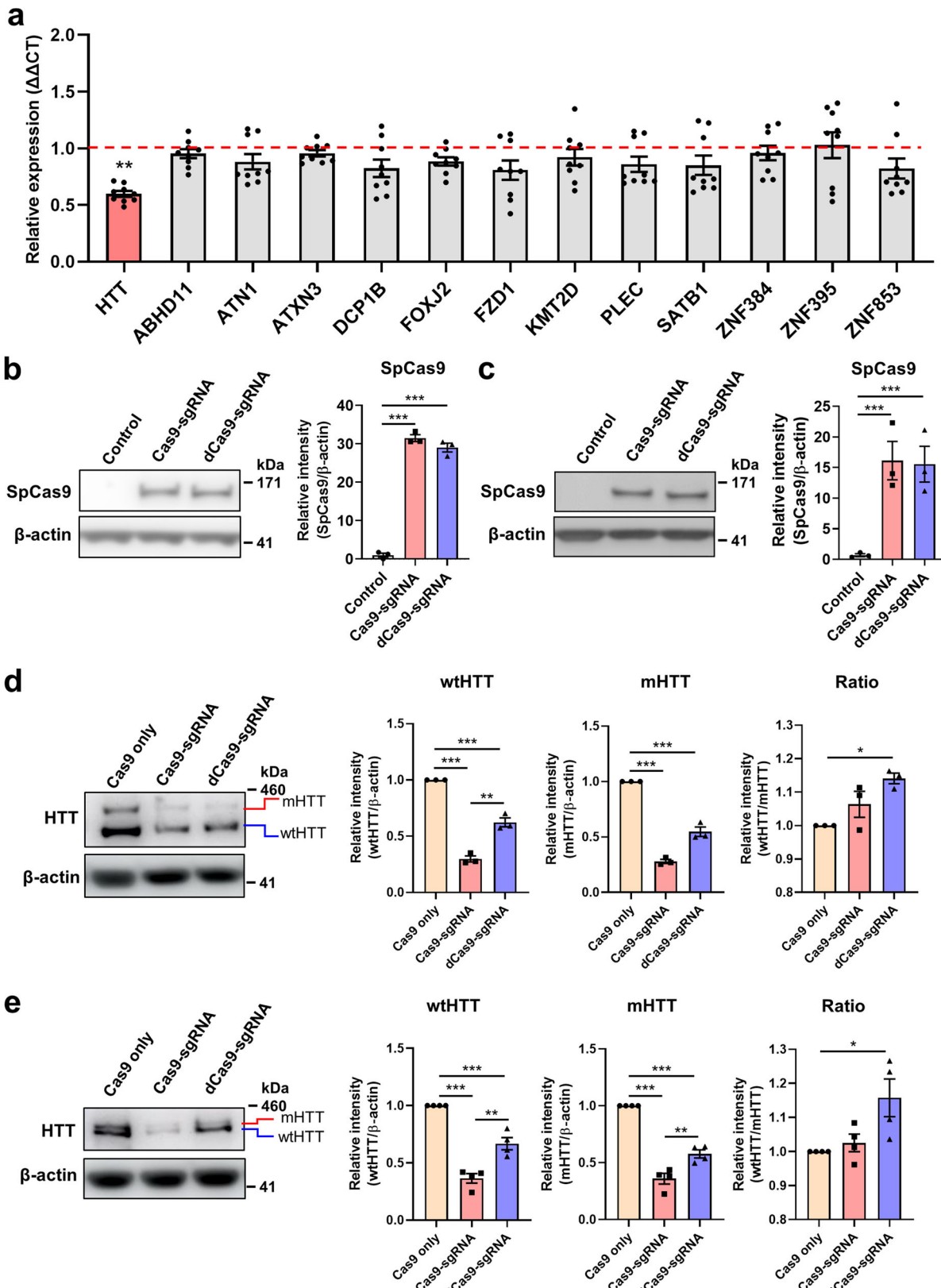

to identify Cas9 system off-target sites that were targeting HTT sites. A high DNA cleavage score indicates that Cas9 nuclease causes double-strand breaks at those locations. When off-target effects were also observed using Digenome-seq. in GM09197 fibroblasts treated with Cas9-sgRNA, the number of in vitro cleavage sites were 648, indicating relatively many off-target sites

(Supplementary Fig. 4a). These results indicated that CRISPR-Cas9 is expected to cause DNA damage due to off-target sites and dCas9 treatment without DSBs will be relatively safe.

We treated HD fibroblasts with Cas9-sgRNA or dCas9-sgRNA, and performed quantitative real-time PCR (qRT-PCR) and Western blot analyses to measure the expression of SpCas9 at

**Fig. 2 CRISPRi suppresses mHTT relative to wtHTT in human HD fibroblasts. a** Transcriptional repression at potential off-target sites identified by Cas-OFFinder in HD fibroblasts (GM09197). While on-target site (in *HTT*) was significantly decreased in results of qRT-PCR, the potential off-target sites showed no significant difference compared to the Cas9-only-treated control group (red dotted line) ($n = 3$, each). Error bars represent mean ± S.E.M. **$P < 0.01$ versus Cas9 only-treated control group, by independent t-test. **b** Representative Western blot images (left panel, respectively). Quantitative analysis of SpCas9 protein abundance indicated that both Cas9 and dCas9 were stably expressed in HD fibroblast lines (GM09197: 21/180 CAG repeats) at 10 days after treatment ($n = 3$, each). **c** Representative Western blot images (left panel, respectively). Quantitative analysis of SpCas9 protein abundance indicated that both Cas9 and dCas9 were stably expressed in HD fibroblast lines (GM21756: 15/70 CAG repeats) at 10 days after treatment ($n = 3$, each). **d** Western blot images showing mHTT and/or wtHTT levels after treatment with Cas9-sgRNA, dCas9-sgRNA, or Cas9 alone in the GM09197 fibroblast (left panel). mHTT protein expression was significantly reduced by both Cas9-sgRNA and dCas9-sgRNA compared with Cas9 alone ($n = 3$, each). Although each treatment decreased levels of both wtHTT and mHTT, the dCas9-sgRNA-treated group had significant relative preservation of wtHTT protein levels. Consequently, the ratio of wtHTT to mHTT was significantly higher in the dCas9-sgRNA-treated group. **e** Representative Western blot images in GM21756 fibroblast (left panel). mHTT protein expression was significantly reduced by both Cas9-sgRNA and dCas9-sgRNA compared with Cas9 alone, and Cas9-sgRNA treatment decreased mHTT expression compared to Cas9-sgRNA treatment ($n = 4$, each). Although each treatment decreased levels of both wtHTT and mHTT, the dCas9-sgRNA-treated group had significant relative preservation of wtHTT protein levels. Consequently, the ratio of wtHTT to mHTT was significantly higher in the dCas9-sgRNA-treated group. Error bars represent the mean ± S.E.M. *$P < 0.05$, **$P < 0.01$, and ***$P < 0.001$; by one-way ANOVA with Bonferroni comparison.

10 days after treatment. The mRNA and protein levels of SpCas9 were stable expressed in both Cas9-sgRNA- and Cas9-sgRNA-treated groups but not in the untreated control group (Fig. 2b, c and Supplementary Fig. 4b–e, h, i).

GM09197 showed a clear difference between the sizes of the normal and mutant proteins whereas GM21756 showed similar sizes between the normal and mutant proteins on western blots using an antibody (clone name, EPR5526) (Supplementary Fig. 3a). 4C8 antibody expressed total HTT protein as one band in western blot (Supplementary Fig. 3b). Expression of wtHTT and mHTT proteins in HD fibroblasts treated with Cas9-sgRNA or dCas9-sgRNA was measured using EPR5526 antibody to distinguish wtHTT and mHTT. Expression of both wtHTT and mHTT was reduced in cells treated with Cas9 or dCas9 compared to the Cas9 only group. In addition, Cas9-treated cells showed a greater reduction in both wtHTT and mHTT expressions compared to dCas9-treated cells in both GM09197 and GM21756 cell lines (Fig. 2d, e). The ratio of wtHTT to mHTT was higher in cells treated with dCas9-sgRNA than in the control, whereas there were no differences between the Cas9-sgRNA and control groups in both GM09197 and GM21756 cell lines (Fig. 2d, e). When total HTT expression was measured using 4C8 antibody in both GM09197 and GM21756 cell lines, total HTT was significantly reduced by Cas9-sgRNA and dCas9-sgRNA treatment (Supplementary Fig. 3c, d). Following the Cas9-sgRNA or dCas9-sgRNA treatment, we observed a significant reduction in mHTT protein levels when an mHTT antibody (clone name, 1C2) was used to detect polyglutamine region in a GM09197 cell line (Supplementary Fig. 3e, f). We provided analysis to difference in mHTT protein versus wtHTT protein for the all groups (supplementary Fig. 3g, h). Unlike the groups treated with Cas9-sgRNA or Cas9 only, the mHTT protein was significantly decreased compared to the wtHTT protein in the group treated with dCas9-sgRNA in both GM09197 and GM21756 fibroblasts. This result suggests that dCas9 has more activity in the mutant allele with longer CAG repeats than in the wild-type allele.

In normal fibroblasts, expression of HTT protein showed a tendency to decrease in the Cas9-sgRNA-treated cells when an EPR5526 antibody was used to detect HTT protein (Supplementary Fig. 4f). The HTT protein expression using 4C8 antibody was ultimately reduced in Cas9- and dCas9-treated groups while its counterpart Cas9 only group did not show any reduction (Supplementary Fig. 4g). Moreover, Cas9-sgRNA and dCas9-sgRNA reduced the HTT protein in HD fibroblasts containing 40–50 CAG (Supplementary Fig. 4j, k). These results showed that the mentioned strategies using sgRNA targeting CAG region can reduce mHTT and wtHTT in both normal fibroblasts and HD fibroblasts.

Because there are more CAG repeats in the mutant allele, it suggests that dCas9-sgRNA targets the mutant allele more than the normal allele as more CAG repeats would lead to more dCas9 binding, thereby preserving expression of wtHTT compared to Cas9-sgRNA in HD fibroblasts (GM09197, GM21756) (Fig. 2d, e).

**CRISPRi delays disease progression in a mouse model of HD.** To examine the effects of the CRISPRi system in vivo, we injected Cas9-sgRNA or dCas9-sgRNA lentivirus into the striatum of 4-week-old R6/2 mice, a model of HD containing approximately 160±5 CAG repeats. The Cas9 only vector that lacked sgRNA was used as a control. The experimental timeline is shown in Fig. 3a. We evaluated behavioral performance using rotarod and open field tests during 8-9 weeks after treatment to determine whether Cas9-sgRNA or dCas9-sgRNA expression affects neurobehavioral function in this HD mouse model. Rotarod tests at constant speeds (12 and 16 rpm) showed that the latency periods in the dCas9-sgRNA group were significantly improved at 6–12 weeks compared with the Cas9-sgRNA, Cas9-only, and PBS groups (Fig. 3b). In addition, although there were no dramatic differences in latency time at accelerating speed, the latency periods in the dCas9-sgRNA group were significantly improved at 6–11 weeks compared with the Cas9-sgRNA, Cas9-only and PBS groups (Supplementary Fig. 5). These results suggest that the constant speed rotarod may be more sensitive to motor function than the accelerated speed rotarod in the CRISPR-Cas9 system treated with HD mice, though acceleration speed may yield specific results. Open field tests showed dramatic recovery in locomotor activity in the dCas9-sgRNA group. However, treatment with Cas9-sgRNA, Cas9 only or PBS in the striatum did not lead to improvements in neurological function in HD mice (Fig. 3c). These results indicated that dCas9-sgRNA treatment delays disease progression in motor function of R6/2 mice.

To provide insight into the effects of the CRISPRi or CRISPR-Cas9 system in mice in the absence of HD, we performed that 4-week-old wild-type mice were received bilateral injection in both striata with Cas9 only, Cas9-sgRNA and dCas9-sgRNA and then we evaluated behavioral performances with rotarod and open field tests (Fig. 3d). Up to 9 weeks after surgery, there were no differences in both rotarod running latency (Fig. 3d) and open field locomotor distance (Fig. 3e) among the groups. This result demonstrates that CRISPRi or CRISPR-Cas9 system has no effect on behavioral performance in wild-type mice and the CRISPRi treatment might be prone to affect HD which has more CAG repeats.

To investigate Cas9-induced molecular side-effects, qRT-PCR and immunohistochemistry were performed in the

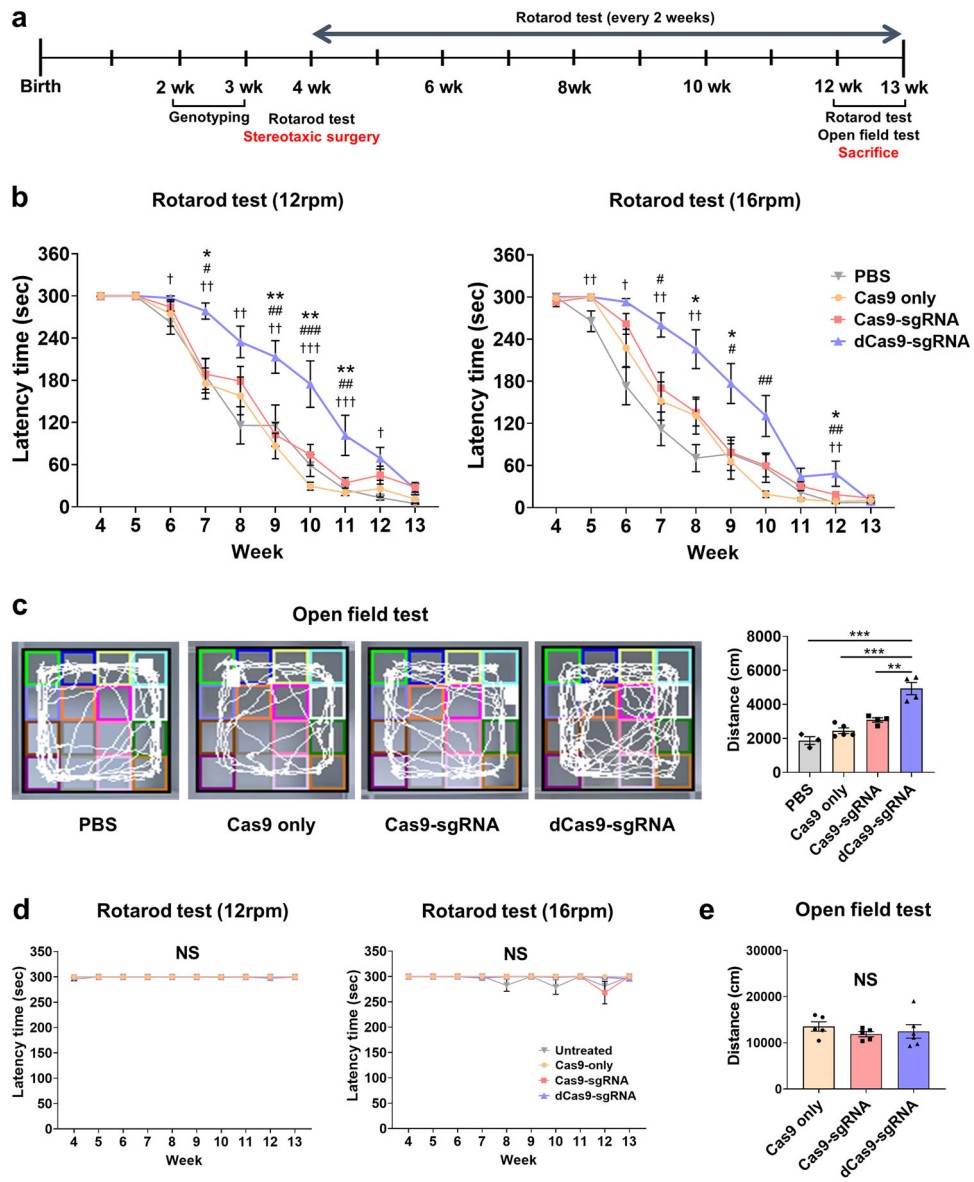

**Fig. 3 CRISPRi delays disease progression in a mouse model of HD. a** The in vivo experimental design. After HD was diagnosed by genotyping at 2–3 weeks of age, mice received Cas9 only, Cas9-sgRNA, or dCas9-sgRNA in their bilateral striata at 4 weeks of age. **b** Results from a rotarod test at constant speed (12 rpm and 16 rpm). When motor performance was evaluated by the rotarod test, dCas9-sgRNA-treated mice ($n = 8$) exhibited delayed motor deterioration compared to Cas9-sgRNA-treated mice ($n = 14$), Cas9 only control group ($n = 8$) and PBS control group ($n = 11$) at 2–8 weeks after treatment (at 6–12 weeks of age). Error bars represent the mean ± S.E.M. *$P < 0.05$ and **$P < 0.01$ versus Cas9-sgRNA; ##$P < 0.01$ and ###$P < 0.001$ versus Cas9 only; †$P < 0.05$, ††$P < 0.01$, and †††$P < 0.001$, versus PBS, by one-way ANOVA with Bonferroni comparison. **c** Open field test. When locomotor activity was evaluated by an open field test, dCas9-sgRNA-treated mice ($n = 4$) showed improved locomotor activity compared to Cas9-sgRNA- ($n = 4$), Cas9 only- ($n = 5$) and PBS- ($n = 3$) treated groups at 13 weeks of age. Error bars represent the mean ± S.E.M. *$P < 0.05$, **$P < 0.01$, and ***$P < 0.001$, by one-way ANOVA with Bonferroni comparison. **d, e** Behavior outcomes in wild-type mice in the absence of HD. **d** When motor performance was evaluated by the rotarod test (12 rpm and 16 rpm), there were no significant differences among the groups after treatment (Untreated, $n = 8$; Cas9 only, $n = 5$; Cas9-sgRNA, $n = 5$; dCas9-sgRNA, $n = 7$). **e** locomotor activity was evaluated by an open field test, there were no significant differences among the groups after treatment (Cas9 only, $n = 5$; Cas9-sgRNA, $n = 5$; dCas9-sgRNA, $n = 6$) at 13 weeks of age. Error bars represent the mean ± S.E.M. NS, not significant ($P > 0.05$). Statistical significance was determined according to one-way ANOVA followed by Bonferroni comparison.

striatum of wild-type mice at 8 weeks of age. *SpCas9* was significantly expressed in both Cas9-sgRNA- and dCas9-sgRNA-treated groups but not in the control group. The *Htt* levels expression was significantly reduced by both Cas9-sgRNA and dCas9-sgRNA compared with Cas9 alone. In addition, to assess the effect on other CAG repeat regions we looked at *Atxn3* levels, expression of *Atxn3* was significantly lower in Cas9-sgRNA-treated wild-type mice than dCas9-sgRNA- and Cas9 only-treated mice (Supplementary Fig. 6a).

When immunohistochemisty was also performed in the striatum of wild-type mice, the densities of β-tubulin+ neurons were significantly higher in dCas9-sgRNA-treated mice versus Cas9-sgRNA-treated and control mice, and the densities of β-tubulin+ neurons decreased in Cas9-sgRNA-treated mice versus dCas9-sgRNA- and Cas9 only-treated mice (Supplementary Fig. 6b). These results suggest that Cas9-sgRNA treatment has greater molecular side-effects than dCas9-sgRNA treatment.

**CRISPRi protects striatal neurons and reduces mHTT against cell death in the brain**. We confirmed that off-target effects were not observed in mRNA-seq results of the striata of dCas9-sgRNA-treated mice relative to that of Cas9 only-treated mice (Supplementary Data 3). When we next performed Digenome-seq in the striata of mice treated with Cas9-sgRNA targeting CAG repeat region, the number of in vitro cleavage sites was 5152, indicating many off-target sites (Supplementary Fig. 7a).

The mRNA levels of *SpCas9*, *mHTT*, phosphatase 1 regulatory subunit 1B (*Ppp1r1b*), D1 dopamine receptor (*Drd1*), and D1 dopamine receptor (*Drd2*) were assessed using qRT-PCR. We confirmed the expression of SpCas9 in the striatum of Cas9-sgRNA- and dCas9-sgRNA-treated HD mice at 13 weeks of age using qRT-PCR (Fig. 4a). When the mRNA levels of *mHTT* was assessed using qRT-PCR, the expression of *mHTT* decreased in mice treated with dCas9-sgRNA and Cas9-sgRNA (Fig. 4a). We also confirmed the expression of *mHTT* gene in the striatum of R6/2 mice 4 weeks after treatment (8 weeks of age). *mHTT* significantly suppressed in both Cas9-sgRNA- and Cas9-sgRNA-treated HD mice compared to control mice (Supplementary Fig. 7b). These results indicated that both dCas9 and Cas9 treatments reduced *mHTT* by sgRNA targeting the CAG repeat region.

In addition, we confirmed the expression of genes which expressed in medium spiny neurons such as *Ppp1r1b*, *Drd1*, and *Drd2*. We found that the *Ppp1r1b* gene, encoding the dopamine- and cAMP-regulated neuronal phosphoprotein (DARPP-32), a GABAergic neuronal marker, was significantly increased in dCas9-sgRNA-treated HD mice compared to Cas9-sgRNA-treated mice. In addition, *Drd1* and *Drd2* which expressed in striatal medium spiny neurons, were significantly increased in the dCas9-treated group (Fig. 4a).

We evaluated levels of DARPP-32, neuronal nuclei (NeuN, a mature neuronal marker), neuron-specific beta-III tubulin (βIII-tubulin, an early neuronal marker), and BDNF using Western blotting or immunohistochemistry in the striatum of R6/2 mice. In Western blots, DARPP-32, NeuN, and BDNF levels were significantly increased in striatal tissue from dCas9-sgRNA-treated mice compared to Cas9-sgRNA- and Cas9 only-treated mice (Fig. 4b). In immunohistochemical analysis, the densities of βIII-tubulin+, NeuN+, and BDNF+ cells were significantly higher in striatal sections from dCas9-sgRNA-treated mice compared to Cas9-sgRNA- and Cas9 only-treated mice (Fig. 4c). The percentage of mHTT+ cells to DAPI+ cells was significantly reduced in dCas9-sgRNA-treated mice compared to Cas9 only-treated mice (Fig. 4c). The reduction of mHTT+/DAPI+ cells in the dCas9-sgRNA group, but not in the Cas9-sgRNA-treated group, can be explained by the relative preservation of striatal DAPI+ cells (Fig. 4c). These data indicate that dCas9-sgRNA treatment could be a useful therapeutic method for reducing mHTT and protecting striatal neurons.

We also performed TUNEL staining of the striatum of HD mice to detect cell death. We found significantly fewer TUNEL+ cells in the striatum of dCas9-sgRNA-treated R6/2 mice compared to Cas9-sgRNA-treated R6/2 mice (Fig. 4d). These results suggest that dCas9-sgRNA treatment can protect striatal neurons against cell death in brains of HD mice, although both dCas9 and Cas9 treatments reduced *mHTT* by sgRNA targeting the CAG repeat region.

The expression of genes, namely, *SpCas9*, *mHTT*, *Atxn3*, *Ppp1r1b*, *Drd1*, and *Drd2*, were also investigated using qRT-PCR in the striatum of R6/2 mice at 4 weeks after treatment. Consequently, *SpCas9* was significantly expressed in both Cas9-sgRNA- and Cas9-sgRNA-treated groups but not in the untreated control group. *mHTT* significantly suppressed in both Cas9-sgRNA- and Cas9-sgRNA-treated HD mice compared to control

mice. *Atxn3* expression showed significantly decrease in Cas9-sgRNA compared to the dCas9-sgRNA-treated groups. The expression of *Ppp1r1b* and *Drd2* was higher in the dCas9-sgRNA-treated group compared to the Cas9-sgRNA-treated group. The result of *Drd1* expression showed a significant decrease in both dCas9-sgRNA-treated and Cas9-sgRNA-treated mice. (Supplementary Fig. 7b). The levels of Darpp-32, NeuN, and BDNF were also investigated using western blotting in the striatum of 8-week-old R6/2 mice. The Darpp-32, NeuN, and BDNF levels were significantly increased in the striatum of dCas9-sgRNA-treated mice compared to Cas9-sgRNA- or Cas9 only-treated mice (Supplementary Fig. 7c). To confirm the expression of Cas9 in vivo, we performed immunohistochemistry in HD mouse brain-treated with lentiviral vector containing CRISPR system. SpCas9+ cells were expressed by dCas9-sgRNA-, Cas9-sgRNA-, or Cas9 only-treatment in R6/2 mice brain at 4 weeks after treatment (Supplementary Fig. 7d). These results suggested that SpCas9 protein was expressed in vivo by CRISPRi or CRISPR-Cas9 at 4 weeks post-treatment.

## Discussion
In this study, we demonstrated that DNA DSB-free CRISPRi including sgRNA targeting the CAG repeat region delays functional deterioration of HD compared to CRISPR-Cas9 nuclease treatment. We propose two explanations for this phenomenon: (1) dCas9 treatment is more effective in retaining wtHTT while decreasing mHTT compared to Cas9, and (2) administration of dCas9 has therapeutic effects such as improved behavioral function and neuroprotection.

HD is a progressive neurodegenerative disease caused by the expression of *mHTT* in which the CAG trinucleotide repeats are expanded at the exon 1 of the *HTT* gene[1,2]. The mHTT causes neuronal dysfunction and degeneration in striatum and cerebral cortex, leading to severe neuropathological changes without fundamental treatment options so far. Therefore, genome editing directly targeting the DNA may be useful for HD as a potential strategy to permanently suppress *mHTT*[30,31].

Genome editing is a promising therapy for HD with a CAG repeat (polyQ-Htt) in the *HTT* gene as a single treatment can permanently repair specific genes. Recent studies using genome editing strategies have showed behavioral improvement in HD mice. R6/2 mice treated with ZFN for allele-specific transcriptional repression improved neurobehavior in open field and clasping tests[16]. CRISPR-Cas9 nuclease-treated R6/2 mice also showed that motor deficits improved and lifespan increaed[17]. In the heterozygous HD140Q-KI mice permanent suppression of endogenous *mHTT* expression using CRISPR-Cas9 nuclease significantly improved behavior test results that reflect motor dysfunction including rotarod, balance beam, and grip strength test were significantly improved[17]. The allele-selective zinc finger protein transcription factors to target mutant CAG repeat region for the treatment of HD showed that transcription repression of *mHTT* and neuropathological improvement in vitro and in vivo[16]. But CRISPR-Cas9 system induce DSBs which can cause cell death, mutagenesis, and off-target effects. Therefore, we applies a treatment strategy using CRISPRi without DSBs for HD.

CRISPR system requires sgRNA to bind target DNA. The target site should contain a PAM sequence, and most commonly use NGG PAM. Despite the lower efficiency compared to NGG, NAG PAM was used to recognize the CAG repeat region. Previous reports demonstrated that NAG triplets are less tolerated in mismatches and that sites with NAG PAMs are recognized with one-fifth the efficiency of target sites with 5′-NGG, the canonical SpCas9 PAM[32,33]. We first designed dCas9 and Cas9 nuclease to bind to the polyQ track (containing CAG PAM). Deep

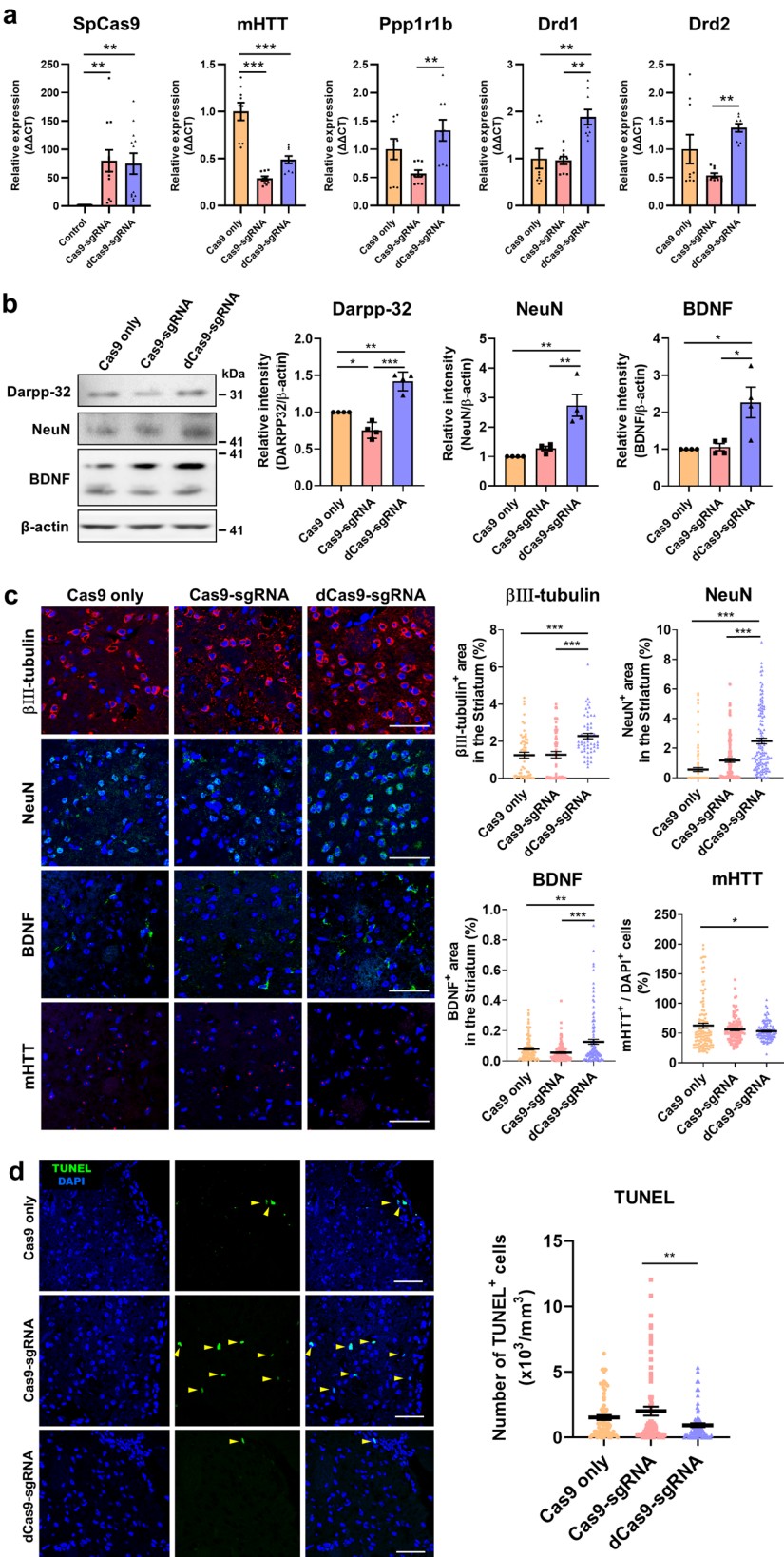

sequencing at 3 days after co-transfection of plasmids including Reporter only, Cas9 only with reporter, Cas9-sgRNA with reporter or dCas9-sgRNA with reporter in HEK293T cells, showed that the number of CAG repeats was reduced only in the Cas9-sgRNA group.

We then applied the CRISPRi system containing catalytically dCas9 on both in vivo and in vitro. To compare the effects of functional nuclease, the CRISPR-Cas9 nuclease group also was additionally investigated. We found that treatment of dCas9 with sgRNA targeting CAG repeats suppressed mHTT relative to

**Fig. 4 CRISPRi preserves striatal neurons in a mouse model of HD. a** mRNA expression of genes of interest using qRT-PCR. SpCas9 was significantly expressed by Cas9-sgRNA and dCas9-sgRNA in the striatum of R6/2 mice at 13 weeks after treatment ($n = 3$, each). mHTT levels expression was significantly reduced by both Cas9-sgRNA and dCas9-sgRNA compared with control ($n = 3$, each). The expression of Ppp1r1b, Drd1, and Drd2 was higher in the dCas9-sgRNA-treated group compared to the Cas9-sgRNA-treated group ($n = 3$, each). **b** Images of Western blots probed with antibodies recognizing DARPP-32, NeuN, and BDNF in striatal tissue from 13-week-old R6/2 mice. β-actin was used as a loading control (left panel). Quantified results from Western blot analysis (right panel). Darpp-32, NeuN and BDNF levels increased versus Cas9-sgRNA-treated or Cas9 only-treated mice ($n = 4$, each). **c** Representative confocal microscopic images of βIII-tubulin[+], NeuN[+], and BDNF[+] cells in striatal sections from R6/2 mice (left panel; scale bars, 50 μm). Quantified results from immunohistochemical analysis (right panel). The densities of βIII-tubulin[+], NeuN[+], and BDNF[+] cells were significantly higher in the striatum from dCas9-sgRNA-treated versus Cas9-sgRNA-treated and control mice (βIII-tubulin, $n = 3$, each; NeuN and BDNF, $n = 4$, each). Error bars represent the mean ± S.E.M. *$P < 0.05$, **$P < 0.01$, and ***$P < 0.001$, by one-way ANOVA with Bonferroni comparison. The percentage of mHTT[+] cells to DAPI[+] cells was significantly reduced in the dCas9-sgRNA-treated mice compared to Cas9 only-treated mice (mHTT, $n = 4$, each). Error bars represent the mean ± S.E.M. *$P < 0.05$, by one-way ANOVA with Tukey comparison. **d** Representative confocal microscopic images of the TUNEL assay performed on the striatum of R6/2 mice (left panel; scale bars, 50 μm), showing that the number of TUNEL[+] cells (yellow arrows) significantly decreased in dCas9-sgRNA-treated versus Cas9-sgRNA-treated mice (right panel; $n = 3$, each). Error bars represent the mean ± S.E.M. **$P < 0.01$, by one-way ANOVA with Bonferroni comparison.

wtHTT in human fibroblasts derived from HD patient (GM09197 and GM21756 fibroblasts) because there are more CAG repeats in the mutant allele. In addition, When the difference between mHTT and wtHTT was analyzed, unlike the groups treated with Cas9-sgRNA or Cas9 only, the mHTT protein was significantly decreased compared to the wtHTT protein in the group treated with dCas9-sgRNA in both GM09197 and GM21756 fibroblasts. These results indicated that the sgRNA is efficient in targeting the CAG repeat region and that dCas9-sgRNA preserves expression wtHTT more than mtHTT compared to Cas9-sgRNA. In normal fibroblasts with 17–24 CAG repeats, protein levels of mHTT and wtHTT were significantly reduced only in Cas9-sgRNA-treated group. In addition, this study showed that Cas9-sgRNA and dCas9-sgRNA reduced mHTT and wtHTT in HD fibroblasts with 40–50 CAG repeats, suggesting that our strategy using sgRNA targeting CAG region can reduce both mHTT and wtHTT in normal human fibroblasts and HD fibroblasts. Consequently, the wild-type CRISPR-Cas9 can be sufficient for genome editing in a short duration and CRISPRi suggests that longer CAG repeats results in greater inhibition of mHTT.

Because wtHTT protein was reduced by CRISPR using sgRNA with CAG PAM, other genes with CAG repeats might have been affected. When we investigated the expression of genes containing CAG repeats such as *ATXN1, ATN1, ATXN3, ATXN7, CACNA1A*, and *TBP* in human HD fibroblasts, the expression of the genes was significantly reduced by Cas9-sgRNA treatment, but not by dCas9-sgRNA treatment. These results suggest that CRISPRi unlikely to significantly regulate expression of the expected off-target genes containing CAG repeats.

This study showed that dCas9 treatment protected striatal neurons contrary to the Cas9 treatment, consequently resulting in delaying motor decline of the R6/2 HD mice, suggesting dCas9-sgRNA targeting the CAG repeat region could be a relatively safe therapeutic strategy for reducing mHTT relative to wtHTT in HD. Additional experiments revealed that wild-type mice that received either Cas9 only, Cas9-sgRNA or dCas9-sgRNA showed no effect on behavioral performances, suggesting that CRISPRi had no therapeutic effects in normal conditions. Although there was no effects on behavioral performance in wild-type mice, βIII-tubulin[+] cells were preserved in the dCas9-sgRNA-treated mice compared to the Cas9-sgRNA-treated mice. These results suggest that Cas9 did not affect behavioral performances in normal conditions, but it had molecular side-effects such as DSBs-induced cell death.

The R6/2 mouse model containing approximately 160±5 CAG repeats is one of the most widely used in HD studies because it displays typical disease phenotypes with early onset, severe motor decline and short lifespan[34,35]. It can be used to study novel

treatment with the advantage of being able to easily confirm therapeutic effect because the disease phenotype appears obvious. However, because the R6/2 model is truncated N-terminus fragment models, it has limitations in studying full-length protein function. In the future, therapeutic effects should be examined in the HD mouse model with a full-length mHTT such as Q111, zQ175, YAC128, or BACHD which exhibit similar neuropathology of the human with HD[36].

Approximately 95% of GABAergic medium spiny neurons, which contain Drd1 and Drd2 receptors, are located in the striatum; HD is caused by selective degeneration and cell death of these neurons as a result of mHTT accumulation[37]. Therefore, GABAergic neuron survival is important in protection against HD. BDNF, a candidate for HD therapy, is a primary neurotrophic factor delivered from the cortex to the striatum through the corticostriatal tract[38,39]. In our results, the brains of dCas9-sgRNA-treated mice showed increased expression of striatal neuronal markers such as *Ppp1r1b*, *Drd1*, and *Drd2*, and neurotrophic factor BDNF compared to Cas9-sgRNA-treated mice. Importantly, in the striatum of dCas9-sgRNA-treated mice, the expression of *mHTT* was significantly reduced compared to Cas9 only-treated mice, and TUNEL[+] cells were significantly reduced compared to Cas9-sgRNA-treated mice. Taken together, these results suggest that *mHTT* inhibition using CRISPRi may be a relatively safe treatment than CRISPR-Cas9 nuclease in that it can reduce the cell death caused by DNA DSBs, consequently delaying functional deterioration.

This study proposes CRISPRi that does not induce DNA DSBs is a potential therapeutic strategy for HD, and thus has the advantage of avoiding the cell death caused by DNA damage compared to other CRISPR-Cas9-based approaches. Although CRISPRi treatment may be less efficient than CRISPR-Cas9 in terms of suppressing gene expression, CRISPRi strategy relatively preserves the expression of *wtHTT* and protects striatal neurons against DNA damage in HD. Here, the sgRNA targeted the CAG repeat region showed that the dCas9 strategy lowered the risk of cell death due to DNA cleavage of pathologically expanded loci compared to CRISPR-Cas9 nuclease. Collectively, our results provide evidence that DNA DSB-free CRISPRi can delay disease progression in HD.

## Methods

**Plasmids.** LentiCRISPRv2 (Addgene; 52961) was purchased from Addgene (Cambridge, MA, USA). For *HTT* gene suppression, LentiCRISPR v2 plasmid with U6 promoter driving sgRNA expression and EF-1 alpha promoter driving the expression of Puromycin-T2A-HA-NLS-dCas9-NLS (or Puromycin-T2A-Flag-NLS-Cas9-NLS) was used. The sgRNA-Cas9 and sgRNA-dCas9 constructs were generated by cloning the CAG repeat-targeting sgRNA sequences into a lenti-CRISPRv2 containing puromycin resistance. The sgRNA-dCas9 construct was

generated by mutating the Cas9 sequence in lentiCRISPRv2 to include D10A and H840A mutations, which result in catalytic inactivation of Cas9. The Cas9 only construct was generated by single CRISPR-Cas9 vector infections. Spacer sequence of sgRNA to target the CAG repeat region showed in Supplementary Table 1.

**Lentivirus production**. HEK293T cells (ATCC, CRL-11268) were cultured in 10-cm tissue culture dishes until they reached 70–80% confluence, after which plasmids that contained the gene of interest (10 μg), psPAX2, and pMD2G (5 μg and 2.5 μg, respectively), were transfected at a ratio of 4:2:1 using Lipofectamine 2000 (Invitrogen, Carlsbad, CA, USA). After 8 h of transfection, the culture medium was exchanged with 12 ml of growth medium. The viral supernatants were collected 48 h after the initial transfection and filtered through a Millex-HV 0.45 μm low-protein-binding membrane (Millipore, Darmstadt, Germany) before being concentrated 1000-fold by ultracentrifugation (Optima XPN-100 Ultracentrifuge, Beckman Coulter) at 4 °C at 25,000 rpm for 2 h. The pellets were resuspended in phosphate-buffered saline (PBS) or growth medium. The titer of lentivirus was measured with Lenti-X p24 Rapid Titer Kit (Clontech, Mountain View, CA) according to the manufacturer's instructions. The estimated infectious titers pf Cas9-only, Cas9-sgRNA, and dCas9-sgRNA viruses were $2.34 \times 10^{10}$, $2.24 \times 10^{10}$, and $2.20 \times 10^{10}$ TU/ml, respectively. In addition, lentiviral vector constructs encoding Cas9-only ($1.16 \times 10^9$ TU/ml), Cas9-sgRNA ($1.28 \times 10^9$ TU/ml), and Cas9-sgRNA ($1.02 \times 10^9$ TU/ml) with CMV promoter were from VectorBuilder (Santa Clara, CA, USA).

**Cell culture**. HEK293T cells (ATCC, CRL-11268) were cultured in Dulbecco's modified Eagle media (DMEM; Hyclone, Irvine, CA, USA) supplemented with 10% fetal bovine serum (FBS; Gibco, Carlsbad, CA, USA) and 1% penicillin/streptomycin (Gibco) in an incubator (37 °C, 5% $CO_2$). The cells were seeded for transfection in six-well plates at $8 \times 10^4$ cells per well. The HEK293T cells were transfected with 2 μg/well (six-well plate) of DNA using Lipofectamine (Invitrogen) for 6 h.

The HD fibroblast lines (GM09197, GM21756, GM04022, and GM04855, Coriell, Camden, NJ), which was derived from humans with HD, and normal fibroblast lines (GM04775 and GM07492, Coriell), were used in this study. The cells were grown in DMEM (Hyclone) supplemented with 15% heat inactivated FBS, 1% penicillin/streptomycin, and 0.1% MEM nonessential amino acids (MEM NEAA; Gibco) at 37 °C with 5% $CO_2$. The cells were seeded for transduction in six-well plates at $5 \times 10^4$ cells per well. Cells were transduced with Cas9-sgRNA, dCas9-sgRNA, or Cas9 only. During transduction, the cationic polymer polybrene (8 μg/ml, Sigma-Aldrich, St. Louis, MO, USA) was added to the viral media (Multiplicity of infection, MOI = 10). The medium was exchanged to remove the virus the day after transduction. To select cells transduced with the lentivirus constructs, which contain a puromycin-resistance gene, we treated the mixture with 2 μg/ml puromycin (Gibco) until the non-transduced cells (control) were completely dead.

**Reporter cloning**. To confirm the activity of sgRNA (CAG PAM), oligonucleotides containing target sites with CAG PAM were synthesized. The sequence of this insert fragment, with the target site shown in bold and the CAG PAM are in italics (forward; AATTC**AGCAGCAGCAGCAGCAGCAG**CAG, reverse; GATCC*CTG*CTGCTGCTGCTGCTGCTGCTG) and annealed in vitro using a thermocycler (95 °C for 5 min and then ramped down to 25 °C at 5 °C per min). The annealed oligonucleotides were ligated into the reporter vectors digested with EcoR1 and BamH1. And to confirm the activity of Cas9-sgRNA for a target site with 27 CAG repeats, 27 CAG repeat sequence was PCR amplified using the PrimeSTAR polymerase (TAKARA, Tokyo, Japan). Next, the amplicons, using an NEBuilderHiFi DNA assembly kit (New England BioLabs GmbH, Frankfurt, Germany), were assembled with the linearized reporter plasmid at 50 °C for 30 min. After incubation, the products were purified using a MEGA quick-spin total fragment DNA purification kit (iNtRON Biotechnology, Korea) and transformed into competent cells.

**Flow cytometry**. Three days after transfection, Adherent HEK293T cells co-transfected with reporter plasmid were trypsinized and resuspended in 2% FBS in PBS. Single-cell suspensions were analyzed using the FACSAria II cell sorter (BD Biosciences, San Jose, CA, USA). Untransfected cells and cells transfected with either mRFP vector alone or eGFP vector alone were used as controls.

**Deep sequencing**. Five days after transfection, Adherent HEK293T cells co-transfected reporter plasmid (27 CAG repeats) were trypsinized and genomic DNA was extracted from pelleted cells using a Wizard genomic DNA purification kit (Promega, Madison, WI, USA). The target sequences within reporter plasmid were PCR amplified using the PrimeSTAR polymerase (TAKARA) for the Cas9-sgRNA mediated cleavage experiment. We used 500 μg of genomic DNA per sample as a template for the first PCR. The PCR products were purified with a MEGAquickspin Total Fragment DNA Purification kit (iNtRON Biotechnology). For the second PCR, 100 ng of purified PCR products from the first amplification were annealed with an Illumina adapter and barcode sequences. The resulting products were

isolated, purified, mixed, and analyzed by paired-end read sequencing using Illumina MiniSeq.

**Off-target analysis**. Potential off-target sites for sgRNA sequences in the human genome were determined by the Cas9-OFFinder algorithm previously described[40]. To determine the relative expression of sequences containing potential off-target sites, human HD fibroblasts were transduced with a lentivirus encoding dCas9-sgRNA cassette. One week after transduction, total RNA was extracted using a RNeasy Mini Kit (Qiagen, Hilden, Germany) and reverse-transcribed using PrimeScript™ RT Master Mix (TAKARA). The gene qRT-PCRs were performed using SYBR Green (Applied Biosystems, Carlsbad, CA, USA) with the primers listed in Supplementary Table 1. The results of on-target sites (*HTT*) and off-target binding sites including *ABHD11, ATN1, ATXN3, DCP1B, FOXJ2, FZD1, KMT2D, PLEC, SATB1, ZNF384, ZNF395,* and *ZNF853* were normalized to *GAPDH* expression levels. Several genes, such as *ATXN1, ATN1, ATXN7, CACNA1A,* and *TBP*, were additionally identified for off-target effects. All experiments were performed in triplicate.

**Ethics approval**. All mice were housed in a facility accredited by the Association for Assessment and Accreditation of Laboratory Animal Care (AAALAC) and were kept under controlled climate conditions with a 12-h light/dark cycle and were provided standard food and water ad libitum. All experimental procedures and animal care were approved by the Yonsei University Health System Institutional Animal Care and Use Committee (YUHS-IACUC approval No: 2015-0396, 2019-0100) and all procedures comply with the IACUC guidelines.

**Mice and stereotaxic injection**. In this study we used a total of thirty R6/2 mice, a transgenic mouse model of HD that expresses a version of exon 1 of the human *HTT* gene that includes 160 ± 5 CAG repeats[35]. For stereotaxic injection, the mice were anesthetized with ketamine (100 mg/kg) and xylazine (10 mg/kg) by intra-peritoneal (i.p.) injection and were randomly assigned to three groups for treatment with a lentiviral construct (Cas9 only, n = 8; Cas9-sgRNA, n = 14; dCas9-sgRNA, n = 8) at postnatal week 4 (P28). The mice received a lentiviral construct with $2 \times 10^6$–$10^7$ TU/μl or PBS (2 μl each) in both striata using stereotaxic coordinates (AP + 0.4 mm from bregma; ML ± 1.9 mm from bregma; DV −3.3 mm from dura). Body temperatures were maintained at 37 °C in a heating chamber until the mice fully awakened.

To provide insight into the effects of the lentiviral construct or Cas9 expression in R6/2 mice, we performed additional experiments as to the matched cohort of PBS group (PBS, n = 11). This study was also to confirm the effects of the lentiviral construct in R6/2 mice at 8 weeks of age: Cas9 only (n = 4), Cas9-sgRNA (n = 4), and dCas9-sgRNA (n = 4). In addition, to investigate that CRISPRi or CRISPR-Cas9 system has any effect on wild-type mice in the absence of HD, we established new cohorts of 25 wild-type mice (12 weeks old): untreated (n = 8), Cas9 only (n = 5), Cas9-sgRNA (n = 5), and dCas9-sgRNA (n = 7); twelve wild-type mice (8 weeks old): Cas9 only (n = 4), Cas9-sgRNA (n = 4), and dCas9-sgRNA (n = 4).

**Rotarod performance**. The rotarod (No. 47600; UGO Basile, Comerio, VA, Italy) test was used to assess motor coordination and balance at constant speed (12 rpm and 16 rpm) and accelerating speed (4–40 rpm). The latency period before mice fell from the rod was measured twice during each test, and if the interval between two recorded scores was more than 60 s, the latency period was measured once more. Individual tests were terminated at a maximum latency of 300 s.

**Open field activity test**. Locomotor activity tracing was performed in an open field arena (30 × 30 × 30 cm). Mice were placed individually into the arena and monitored with a video camera. The open field box was cleaned with 70% ethyl alcohol and air-dried prior to the commencement of each trial for every mouse. The resulting data were analyzed using the video tracking system Smart Vision 2.5.21 (Panlab, Barcelona, Spain). In the tracking system, the floor of the open field is divided into 16 spaces (4 × 4). The distance of movement was assessed for 5 min.

**RNA extraction**. Total RNA was extracted from the cells and striata of mouse brains using Trizol (Invitrogen Life Technologies, Carlsbad, CA, USA) according to the manufacturers' protocols. Extracted RNA was quantified RNA concentrations with an A260/A280 ratio for quality control analysis using a NanoDrop 2000 (Thermo Fisher Scientific).

**Quantitative real-time PCR (qRT-PCR)**. cDNA synthesis was performed using Prime Script™ Reverse Transcriptase (TAKARA). Quantitative PCR reactions were prepared with SYBR Green (Invitrogen) according to the manufacturer's protocol. Reactions were run on a Light Cycler thermal cycler (Roche Diagnostics, Mannheim, Germany) with the oligonucleotide primers reported in Supplementary Table 1. The results are expressed as fold-increase in mRNA expression of the gene of interest normalized to *Gapdh* (or *GAPDH*) expression by the ΔΔCt method. All experiments were performed in triplicate.

**mRNA sequencing**. RNA sequencing was processed for transcriptome analysis by Macrogen, Inc. (Seoul, Korea) as previously described[41,42]. The extracted total RNA of mouse brain samples was transcribed into a library of templates that can suitable for cluster generation using the TruSeq RNA Sample Preparation Kit (Illumina). Next, the transcriptome analysis was performed by follow procedures: Total 1000 ng RNA were selective and purified primary mRNA (poly-adenylated RNA) using magnetic beads to conjugate oligo-dT. In order to convert this mRNA into single-stranded cDNA, reverse transcriptase and random hexamer primers were used, and DNA-dependent synthesis of the second strand was suppressed by adding Actinomycin D. After removing the RNA template, double-stranded cDNA was synthesized by adding deoxyribolidine triphosphate (dUTP) instead of deoxythymidine triphosphate (dTTP). And then, a single A base was added on the 3′ end of the adapter with a single T base overhang, and the amount of the sequence-ready library was amplified by polymerase chain reaction using adapter-ligation cDNA. During this amplification process, the polymerase stalls when it encounters a U base, the second strand becomes a bad template, and the amplified material retain information through one strand. The final cDNA library was quantified by qPCR (Kapa Library Quant Kit; Kapa Biosystems, Wilmington, MA) and normalized to 2 nmol/L for sequencing.

**Digenome sequencing**. As presented in Kim et al., intact genomic DNA and Cas9-digested genomic DNA was sequenced to depth of 30x to 40x by Macrogen, Inc. (Seoul, Korea) using whole-genome sequencing (WGS)[43]. The WGS data were used to assign a DNA cleavage score to each nucleotide position in the entire genome as previously described[39] and the DNA cleavage site was identified using the Digenome provided at https://github.com/chizksh/digenome-toolkit2.

**Western blotting**. Harvested cells or mouse brain tissues were dissolved in RIPA lysis buffer (Thermo Fisher Scientific, Waltham, MA, USA) containing protease inhibitor cocktail (Cell Signaling Technology, Beverly, MA, USA) and incubated on ice for 20 min. Cell lysates were centrifuged at $13,000 \times g$ for 15 min at 4 °C. Total protein concentrations in the supernatants were determined using a Bradford protein assay kit (Thermo Fisher Scientific). Proteins (30 μg) were solubilized in NuPAGE® LDS sample buffer (Invitrogen), denatured for 10 min at 90 °C, and loaded into NuPAGE®4–12% Bis-Tris gels and run in 1x NuPAGE® MES SDS running buffer (Invitrogen) at 120 V for 2 h. The resolved proteins were then transferred onto a 0.45 or 0.2 μm Invitrolon™ polyvinylidene difluoride (PVDF; Invitrogen) membrane in 1× NuPAGE® Transfer buffer with 10% (vol/vol) methanol (EMSURE®) using an XCell IITM Blot Module (Invitrogen) on ice for 1 h 20 min. In case of huge size proteins containing wtHTT and mHTT, proteins were subjected to electrophoresis on a NuPAGE®3–8% Tris-Acetate protein gels in 1x Tris-Acetate SDS Running buffer (Invitrogen) and the membranes transferred in 1x Transfer buffer overnight in the cold room (4 °C). The membranes were blocked with 5% bovine serum albumin (BSA) in 1x TBST [Tris-buffered saline (10 Mm Tris-HCl, pH 7.5, 150 mM NaCl) containing 0.05% Tween 20] at room temperature for 1 h and washed three times with 1x TBST. Blocked membranes were incubated at 4 °C overnight with 5% BSA in 1x TBST and the appropriate primary antibodies listed in Supplementary Table 2. The blots were then washed three times with 1x TBST and incubated with horseradish peroxidase-conjugated goat anti-mouse or anti-rabbit secondary antibodies (Santa Cruz Biotechnology, Santa Cruz, CA) at room temperature for 1 h. After washing three times with 1x TBST, immunoreactive bands developed with West-Q Pico ECL Solution (GeneDEPOT, Barker, TX, USA) using the Image Quant LAS-4000 digital imaging system (GE Healthcare, Buckinghamshire, UK). The intensities of the bands were quantified using Multi-Gauge version 3.0 (Fujifilm, Tokyo, Japan). All uncropped Western blot images are presented in Supplementary Fig. 8.

**Immunohistochemistry**. Animals were euthanized, and slowly perfused with 0.9% normal saline and thereafter the same procedure with 4% paraformaldehyde (PFA). The harvested brain tissues were freezed in frozen section compound (Leica, Wetzlar, Germany) using isopentane and cryosectioned at 16-μm thickness using cryomicrotome (Cryostat Leica 1860, Leica biosystem, MI, Italy). The immuno-histochemistry staining was performed on three or four sections over a range of 128 μm. The sections were stained with primary antibodies reported in Supple-mentary Table 2, as well as secondary antibodies such as Alexa Fluor® 594 goat anti-Rabbit of anti-Mouse (1:400), and Alexa Fluor® 488 goat anti-Rabbit or anti-Mouse (1:400, Invitrogen). The sections were mounted on glass slides with a fluorescent mounting medium containing 4′,6-diamidino-2-phenylindole (DAPI; Vectashield, Vector, Burlingame, CA, USA). The stained sections were analyzed using confocal microscopy (LSM700, Zeiss, Gottingen, Germany) and the density was evaluated using ZEN Imaging Software (Blue edition, Zeiss).

**TUNEL assays**. For analysis of apoptosis, mouse brain sections were labeled for apoptotic cells by a Fluorometric TUNEL staining according to the manufacturers' protocols and the selected areas of the striatum were analyzed in four field per tissue. The stained slides containing the cells, and mouse brain sections were analyzed using confocal microscopy (LSM700, Zeiss), and the number of TUNEL+

cells were counted in the fields and the density of TUNEL+ cells in mouse brain sections was evaluated using ZEN Imaging Software (Blue edition, Zeiss).

**Statistics and reproducibility**. All data were expressed as means ± S.E.M. The variables were compared between groups by an independent t-test and among groups by a one-way analysis of variance (ANOVA) with a post hoc Bonferroni or Tukey comparison using SPSS software (IBM Corporation, Armonk, NY, USA; version 25.0). A $P$-value <0.05 was considered statistically significant. In vitro experiments were performed at least three times to confirm reproducibility. Both female and male were used in this study and randomly assigned to the in vivo experimental groups. Mice samples were analyzed by molecular study such as qRT-PCR and western blot using data from three independent experiments.

**Reporting summary**. Further information on research design is available in the Nature Portfolio Reporting Summary linked to this article.

## Data availability

The datasets used and/or analyzed during the current study are available from the corresponding author on reasonable request. Sequencing data of Digenome-seq was deposited in the NCBI Sequence Read Archive (SRA) database with BioProject accession code PRJNA936924 and PRJNA936925. RNA sequencing data from mouse brain have been deposited in SRA under accession code PRJNA645276. All the source data presented in this study are included in the Supplementary Data 2 file. Uncropped scans are provided in Supplementary Fig. 8.

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

## Acknowledgements

The authors thank Medical Illustration & Design, part of the Medical Research Support Services of Yonsei University College of Medicine, for all artistic support related to this work. This study was supported by a grant from the Korean Health Technology R&D Project through the Korea Health Industry Development Institute (KHIDI), funded by the Ministry of Health & Welfare, Republic of Korea (grant no. HI21C1314, HI22C1588 to S.-R.C.), the Korean Fund for Regenerative Medicine (KFRM) grant funded by the Korea government (the Ministry of Science and ICT, the Ministry of Health & Welfare) (grant no. 21A0202L1, 21C0715L1 to S.-R.C.), and Basic Science Research Program through the National Research Foundation of Korea funded by the Ministry of Education (grant no. 2021R1A6A3A01087502 to J.H.Seo).

## Author contributions

J.H.Seo performed most of the experiments, analyzed the data, wrote the manuscript, and designed the study. J.H.Shin performed experiments related to CRISPR-Cas9, analyzed next-generation sequencing results, and conceived the study design. Junwon L. contributed to the experiments and conceived the study design. D.K. and H.-Y.H. analyzed Digenome sequencing. B.-G.N. managed animals and performed the mouse behavioral tests. Jinu L. contributed to the lentivirus production and the development of this study. H.K. interpreted the results. S.-R.C. interpreted the results, wrote the paper, and supervised the project. All authors read and approved the final manuscript.

## Competing interests

The authors declare no competing interests.
