## [Peer Review File · Communications Biology]

Reviewers' comments:

Reviewer #1 (Remarks to the Author):

This manuscript describes the development of a strategy involving CRISPR interference to selectively silence the mutant HTT (mHTT) allele using a dCas9 protein with a CAG-targeting sgRNA. The authors provide data in reporter cells, in primary cells and in an animal model demonstrating various extents of efficacy. However, I have several concerns about the interpretation of the results and the study design that decrease my enthusiasm for the work.

First and foremost, although intended for the mHTT allele, the authors still appear to observe a 30-40% knockdown in WT HTT expression in Figure 2. They also only analyzed allele-specific targeting in cells with patients with very long CAG repeats (70 and 180), which, because of its length, in my opinion biases the study toward more effective repeat silencing (as more repeats would lead to more dCas9 binding and stronger transcriptional interference). I wonder what the degree of mHTT-specific lowering would be in background with 40-50 repeats, as the median repeat length in patients is closer to 44.

There is no convincing data that dCas9 is lowering mHTT in vivo. RT-qPCR or western blot analyses appear to be missing for any time-point. It is thus difficult to establish the link between behavior improvement and the activity of the dCas9 protein.

Additional major comments:

1. The authors state in the abstract and in the manuscript that "CRISPRi is a potential therapy for HD that can outperform CRISPR-Cas9 nuclease in alleviating disease progression and reducing cell death."

I do not believe this is a fair comparison. First, there is no in vivo presented that dCas9 can lower mHTT in the brain. Second, the study does not include the genome-wide or transcriptome-wide measurements needed to most effectively compare these two strategies, and third, in my opinion it is unfair to compare dCas9 to Cas9 for targeting a CAG repeat with a non-canonical PAM. A more effective comparison would involve optimized approaches of both techniques against each other. As that is not possible for this manuscript, the authors should use more moderated language.

2. The targeting strategy is unclear. The authors relied on a CAG PAM for SpCas9, but it's well known that SpCas9 does not tolerate CAG PAMs all that well (<https://www.nature.com/articles/s41467-020-20650-x.pdf> and <https://www.nature.com/articles/srep05405> for example). The authors should better highlight that their strategy relies on a non-canonical PAM and the implications of this for efficient target engagement.

3. The authors' conclusions from the data in Figure 1 seem to be burying a lead to me. The authors found that targeting the repeat with a Cas9 nuclease could reduce the number of repeats, which is potentially a valuable finding (though complicated by the fact that other CAG repeats in the genome would also be targeted). Perhaps the authors should expand or re-frame these results.

4. How were the OT sites in Fig. 2B identified? Were OT effects observed the Cas9 nuclease?

5. For 1E and 1F, it is unclear if there is a significant difference in the ratio of WT/mHTT for the dCas9 group compared to the Cas9 group.

6. Considering the Cas9 group was able to lower mHTT protein even more effectively compared to the

dCas9 group in vitro in 1E and 1F, it is odd that it was unable to improve deficits in the R6/2 model, whereas dCas9 was. qPCR or western blot data is needed to show that it was not able to lower the mHTT protein (which would be unexpected given it lowered mHTT with a 170 CAG repeat in Figure 2).

7. Pg 5, "This result demonstrates that CRISPRi or CRISPR-Cas9 system has no effect on wild-type mice and the CRISPRi treatment has disease-specific effects." The findings showed only no observable side-effects. It is possible, and perhaps likely given that a CAG repeat was targeted, that Cas9 induced molecular side-effects.

8. There is no convincing data demonstrating mHTT knockdown in R6/2 mice (a mHTT to DAPI ratio at end-stage is not convincing). Without this, it is not possible to link the improvements in deficits to a decrease in mHTT. This is a major limitation of the study.

9. Additionally, there is no measurement for off-target effects from the injected tissue. The authors approach involves targeting a CAG repeat and thus has strong potential to influence the expression of other genes.

Minor comments:

1. The statement, "RNAi has some technical limitations related to the volume of solution required for therapy and the delivery method." The authors should elaborate on what they mean by this. Additionally, the same limitation would very well apply to the CRISPRi approach described by the authors, as both RNAi and Cas9 are being investigated for delivery by similar AAV vectors.

2. "RNAi has some technical limitations related to the volume of solution required for therapy and the delivery method, whereas ASOs reduce wtHTT as well as mHTT expression. Moreover, due to RNA undergoing RNase-mediated degradation, these approaches require repeated treatments for long-term effects. In contrast, genome editing using just a single treatment of CRISPR-Cas9 can lead to permanent repair of specific genes."

Two thoughts related to these two sentences. One, it implies RNAi requires repeated delivery, which is not true, as these antisense constructs can be delivered by a viral vector. And two, it brings to the readers attention permanent repair by Cas9, which is not the focus of the manuscript. I encourage the authors to be more precise in their language in the lead-up language to their approach and their results.

Reviewer #2 (Remarks to the Author):

Major:

1. Abstract: "The CRISPRi resulted in the reduced expression of mHTT with relative preservation of the wild-type HTT in human HD fibroblasts and HD mice" – this sentence should be corrected to precisely describe the results of the study; the preservation of the wt HTT in fibroblasts was not proven by the data provided; HTT expression has not been studied in HD mice (WB or RT-qPCR).

2. Introduction: Allele-selective approaches (ASO, RNAi) targeting CAG repeats should be described;

3. "Delivery of dCas9 with sgRNA targeting the CAG repeat region does not damage the targeted DNA" – unclear, please explain why the authors decided to confirm the known fact that dCas9 does not cut DNA? Why did the authors not analyze the instability of the endogenous CAG tract, since the gRNA targets the CAG repeats?

4. "CRISPRi targeting the CAG repeat region decreases mHTT expression while preserving wtHTT expression". The main problem with these results is the poor quality of western blots, which in my

opinion does not allow to measure the wtHTT/mutHTT ratio (see Fig. 2e, and above all Fig. 2f for which only total HTT level should be determined). In addition, it would be very interesting to know how the number of repeats affects the inhibition.

I am not sure if the proof is convincing enough to talk about some preferences in inhibition. Many studies have shown that it is possible to separate mut and wt HTT proteins from HD fibroblasts, even with a smaller difference between the number of glutamins (e.g., publications from D. Corey's lab). Therefore, better quality WB should be demonstrated. Additional WB analysis on normal fibroblast may confirm wt allele preservation. Please provide all the Wblots in the supplementary figures.

5.Line 95, please explain the off-target analysis in more detail; how were the candidates selected? How long are the CAG repeats in these genes? The list does not include e.g., ATXN3 gene, which contains ~23CAG in normal population. Additional evidence is needed to support the conclusion that the presented strategy does not generate off-target effects.

6.It is not clear why, in the murine model, HTT protein and / or transcript levels were not directly tested to confirm the molecular effects.

7.Lines 214, 215 – although in the graph presented in Fig. 4c we can see significantly (*P<0.05) lower level of HTT compared to the control, it should be kept in mind that the staining methods should not be used alone for evaluation of gene expression, but as accompanying more quantitative methods.

8.Please discuss the possible mechanisms underlying the observed effects.

9.Generally there are many sentences with incorrect references, e.g., line 21 (ref. 1,2); line 23 (ref. 4,5); lines 36, 37, 39,... I suggest to cite the original works and not the reviews (e.g. line 37). Please check all the citations in the text.

Minor points:

1.Please add information about virus MOI used in cell experiments

2.Line 114, indicate the CAG tract length in the R6/2 mouse model

3.Line 188, to be more specific - CAG tract rather than polyQ (PROTEIN)

4.Line 274, 314, virus MOI is not specified

5.Line 319, CAG repeat number should be specified

6.Some important references are missing e.g., those concerning allele-selective CAG targeting by ASO/RNAi; different CRISPR-Cas9-based approaches toward HD,

7.Fig.2b, legend, line 529 – the legend that explains this experiment is missing

8.It is not clear what is the difference between Fig.2c and d.

9.Fig. S2b – please specify the cell line

Reviewer #3 (Remarks to the Author):

This publication looks to investigate the use of CRISPRi as a treatment for HD in human cell models and HD mouse models. They assessed behavioural phenotypes and neuronal survival in dCas9-treated mice. Motor function was assessed using open field test and rotarod, both showing an improvement with the dCas9-gRNA- though motor decline still occurred at a substantial rate. These findings suggest that CRISPRi may be used to delay disease onset or progression, which would be valuable in the context of human disease.

I think that this manuscript is important for the future utility of CRISPRi or genome-editing in the context of treatment for disease. The manuscript may benefit from a thorough explanation of experimental design and strategy and a more well-rounded discussion, discussing the translation of CRISPRi as a potential therapeutic treatment in humans.

My major criticism is the use of the R6/2 mouse model which is transgenic for expanded human exon 1 of mHTT only, I think this methodology only really shows its translational benefit if also tested in a full-length knock in model such as the Q111 or zQ175 mouse or even the YAC 128 or BACHD mice. Those are the mouse models the field really focuses on now for therapeutic translation potential. This

paper did not convince me of CRISPRi as a potential novel therapeutic for Huntington's disease.

Other points:

Introduction:

1. I think it would be useful for the reader in the introduction if the authors explained how CRISPRi works. The introduction describes other techniques but does not provide much insight on the reasoning behind CRISPRi or the proposed mechanism of action. Perhaps including references to other diseases where CRISPRi has been utilised or proved effective. There is reference to the desired knockdown of mHTT only, though it's not clear how this is selectively achieved with CRISPRi.

2. There are a number of sentence errors in the introduction;

- line 27; 'there has'

- lines 27-29; slight rewording perhaps

- line 45; doesn't read well, perhaps remove the word 'that'

- sections could be written to flow slightly better, the penultimate paragraph feels like a collection of statements rather than a congruous section.

3. Line 39 mentions ASOs though it should be noted that there are ASOs that reduce both wtHTT/mHTT (Tominerson) and allele-selective ASOs that have been used in clinical trials (Wave).

Results:

4. Figure 1: I am not convinced that Figure 1a schematic accurately explains the system design. It perhaps needs an accompanying explanation that this is due to more Cas9 complexes binding to the elongated polyQ stretch- if this is the proposed way in which it preferentially reduces mHTT.

Figure 1 is titled 'The sgRNA with CAG as PAM sequence reduces the number of CAG repeats'. With what we know about repeat instability and reducing the number of CAG repeats being seen as beneficial, this could perhaps be reworded to indicate that in this instance- this was not the desired effect. The manuscript stance focuses on the efficacy of dCas9, so this should be the focus of the figure- rather than what the Cas9-sgRNA does. "dCas9-sgRNA aims to reduce mHTT expression without altering CAG repeat length"

5. Following on from point 4- provide more detail in the results section as to how the sgRNA preferentially targets the mHTT allele rather than the wtHTT allele.

6. Figure 2: Shows Cas9 expression in fibroblasts 10 days after treatment. Shows HTT KD in two fibroblast lines. Compares KD ratios in HTT and mHTT. I think they interpret this data correctly, though would be good to see full blots in supplementary.

7. Line 102-104: In addition, dCas9-treated cells showed that wtHTT expression significantly increased compared to Cas9-treated cells in both 103 GM09197 and GM21756 cell lines (Fig. 2e, f)- this sentence is not correct, wtHTT was still reduced in dCas9-treated cells, Cas9-treated cells just showed a greater reduction in wtHTT and mHTT expression.

8. Lines 108-110 suggest that findings from supplementary figure 2b indicate greater knockdown of mHTT than wtHTT. Supplementary figure 2b does indeed show reduction of mHTT in dCas9-sgRNA treated cells, however the use of antibody 1C2 which only probes mHTT means that we cannot infer these findings from supplementary figure 2b. This statement should be made in reference to Figure 2e and f, where antibodies probing both alleles have been used.

9. Figure 3: The authors show that dCas9-treated animals perform better on the rotorod at various time points up to 13 weeks. They also show increased exploration in dCas9-treated animals in the open field test. I think the author should make clear when they say 'ameliorate' whether they mean a delay in decline or a change in the rate of decline. One would suggest that treatment delays initial motor symptoms, whereas the other would suggest a consistent benefit to phenotype throughout the

mouses life. Whilst there is an improvement of motor function on the rotorod, animals in all groups show severe motor decline- so I do not think you can report the amelioration of disease progression, alleviation perhaps. It would be good to include a brief description of R6/2 model of HD and its severity.

Response to Reviewers' Comments

Manuscript ID COMMSBIO-22-0171-T, entitled "DNA double-strand break-free CRISPR interference alleviates Huntington's disease progression"

Reviewer #1 (Remarks to the Author):

This manuscript describes the development of a strategy involving CRISPR interference to selectively silence the mutant HTT (mHTT) allele using a dCas9 protein with a CAG-targeting sgRNA. The authors provide data in reporter cells, in primary cells and in an animal model demonstrating various extents of efficacy. However, I have several concerns about the interpretation of the results and the study design that decrease my enthusiasm for the work.

First and foremost, although intended for the mHTT allele, the authors still appear to observe a 30-40% knockdown in WT HTT expression in Figure 2. They also only analyzed allele-specific targeting in cells with patients with very long CAG repeats (70 and 180), which, because of its length, in my opinion biases the study toward more effective repeat silencing (as more repeats would lead to more dCas9 binding and stronger transcriptional interference). I wonder what the degree of mHTT-specific lowering would be in background with 40-50 repeats, as the median repeat length in patients is closer to 44.

Response: We greatly appreciate your careful review. We agree with your comments.

We performed additional experiments in human HD fibroblasts with 40-50 CAG repeats (GM04022, 44/18 CAG repeats; GM04855, 48/20 CAG repeats) and human control fibroblasts (GM04775, 24/17 CAG repeats; GM07492, 21/18 CAG repeats).

We described the additional results as follows (Results, page 6, lines 155-166):

In normal fibroblasts, expression of HTT protein showed a tendency to decrease in the Cas9-sgRNA-treated cells when an EPR5526 antibody was used to detect HTT protein (Supplementary Fig. 3f). The HTT protein expression using 4C8 antibody was ultimately reduced in Cas9- and dCas9-treated groups while its counterpart Cas9 only group did not show any reduction (Supplementary Fig. 3g). Moreover, Cas9-sgRNA and dCas9-sgRNA reduced the HTT protein in HD fibroblasts containing 40-50 CAG (Supplementary Fig. 3j, k). These results showed that the mentioned strategies using sgRNA targeting CAG region can reduce mHTT and wtHTT in both normal fibroblasts and HD fibroblasts.

Because there are more CAG repeats in the mutant allele, it suggests that dCas9-sgRNA targets the mutant allele more than the normal allele as more CAG repeats would lead to more dCas9 binding, thereby preserving expression of wtHTT compared to Cas9-sgRNA in HD fibroblasts (GM09197, GM21756).

There is no convincing data that dCas9 is lowering mHTT in vivo. RT-qPCR or western blot analyses appear to be missing for any time-point. It is thus difficult to establish the link between behavior improvement and the activity of the dCas9 protein.

Response: As you recommended, we added results of RT-qPCR analysis in *mHTT* (human specific-mutant HTT) in the striatum of mouse brain at 8 weeks and 13 weeks of age.

When the mRNA levels of *mHTT* was assessed using qRT-PCR, the expression of *mHTT* decreased in mice treated with dCas9-sgRNA and Cas9-sgRNA (13 weeks of age) (Fig. 4a). We also confirmed the expression of *mHTT* gene in the striatum of R6/2 mice 4 weeks after treatment (8 weeks of age). *mHTT* significantly suppressed in both Cas9-sgRNA- and Cas9-sgRNA-treated HD mice compared to control mice (Supplementary Fig. 5b). These results indicated that both dCas9 and Cas9 treatments reduced *mHTT* by sgRNA targeting the CAG repeat region (Results, page 7, lines 200-207).

Additional major comments:

1. The authors state in the abstract and in the manuscript that “CRISPRi is a potential therapy for HD that can outperform CRISPR-Cas9 nuclease in alleviating disease progression and reducing cell death.”

I do not believe this is a fair comparison. First, there is no in vivo presented that dCas9 can lower mHTT in the brain. Second, the study does not include the genome-wide or transcriptome-wide measurements needed to most effectively compare these two strategies, and third, in my opinion it is unfair to compare dCas9 to Cas9 for targeting a CAG repeat with a non-canonical PAM. A more effective comparison would involve optimized approaches of both techniques against each other. As that is not possible for this manuscript, the authors should use more moderated language.

Response: We have performed additional experiments according to your comments.

To demonstrate that dCas9 can lower *mHTT in vivo*, we have performed RT-qPCR studies in the striatum of R6/2 mice as mentioned above (Fig. 4a and Supplementary Fig. 5b) (Results, page 7, lines 200-207).

Additionally, we performed mRNA-seq and Digenome-seq analysis in the brain of mice.

We confirmed that off-target effects were not observed in mRNA-seq results of the striata of dCas9-sgRNA-treated mice relative to that of Cas9 only-treated mice (Table S3). When we next performed Digenome-seq in the striata of mice treated with Cas9-sgRNA targeting CAG repeat region, the number of in vitro cleavage sites was 5152, indicating many off-target sites (Supplementary Fig. 5a) (Results, page 8, lines 232-236).

2. The targeting strategy is unclear. The authors relied on a CAG PAM for SpCas9, but it's well known that SpCas9 does not tolerate CAG PAMs all that well (<https://www.nature.com/articles/s41467-020-20650-x.pdf> and <https://www.nature.com/articles/srep05405> for example). The authors should better highlight that their strategy relies on a non-canonical PAM and the implications of this for efficient target engagement.

Response: As you pointed out, previous studies have shown that non-canonical PAMs including NAG PAM lesser interact to Cas9 than NGG PAMs. We added the sentence to better highlight our strategy as follows (Introduction, page 3, lines 69-73):

CRISPR-Cas9 system requires single guide RNA (sgRNA) to bind target DNA. The target sites should include a PAM sequence, whereby the most commonly used PAM is NGG. Despite the lower efficiency compared to NGG, various PAMs including NAG, NGA, and NCC have been experimented. Therefore, sgRNA containing NAG PAM sequence for targeting CAG repeat region can be used to treat HD.

3. The authors conclusions from the data in Figure 1 seems to be burying a lead to me. The authors found that targeting the repeat with a Cas9 nuclease could reduce the number of repeats, which is potentially a valuable finding (though complicated by the fact that other CAG repeats in the genome would also be targeted). Perhaps the authors should expand or re-frame these results.

Response: As you suggested, we reframed the Results as follows (Page 4, lines 107-109):

Therefore, dCas9 with sgRNA targeting the CAG repeat region circumvents the risk of altering the DNA sequence, a side effect of administering Cas9 nuclease as complicated by the fact that other CAG repeats in the genome would also be targeted.

4. How were the OT sites in Fig. 2B identified? Were OT effects observed the Cas9 nuclease?

Response: We searched the potential off-target sites via Cas9-OFFinder algorithm. When we identified mRNA expression of sorted genes, off-target effects in *ABHD1*, *ATN1*, *ATXN3*, *DCP1B*, *KMT2D*, *FOXJ2*, *FZD1M*, *PLEC*, *SATB1*, *ZNF384*, *ZNF395*, and *ZNF853* were not observed in GM09197 fibroblasts treated with dCas9-sgRNA (Fig. 2a). When off-target effects were also observed using Digenome-seq in GM09197 fibroblasts treated with Cas9-sgRNA, the number of *in vitro* cleavage sites were 648, indicating relatively many off-target sites (Supplementary Fig. 3a) (Results, page 5, lines 128-134).

5. For 1E and 1F, it is unclear if there is a significant difference in the ratio of WT/mHTT for the dCas9 group compared to the Cas9 group.

Response: As you suggested, we provided images of better quality for Figure 2d and 2e (wtHTT and mHTT). We also revised the sentence as follows (Results, page 5, lines 136-143):

Expression of wtHTT and mHTT proteins in HD fibroblasts treated with Cas9-sgRNA or dCas9-sgRNA was measured using an antibody (clone name, EPR5526) to distinguish wtHTT and mHTT. Expression of both wtHTT and mHTT was reduced in cells treated with Cas9 or dCas9 compared to the Cas9 only group. In addition, Cas9-treated cells showed a greater reduction in both wtHTT and mHTT expressions compared to Cas9-treated cells in both GM09197 and GM21756 cell lines (Fig. 2d, e). The ratio of wtHTT to mHTT was higher in cells treated with dCas9-sgRNA than in the control, whereas there were no differences between the Cas9-sgRNA and control groups in both GM09197 and GM21756 cell lines (Fig. 2d, e).

6. Considering the Cas9 group was able to lower mHTT protein even more effectively compared to the dCas9 group in vitro in 1E and 1F, it is odd that it was unable to improve deficits in the R6/2 model, whereas dCas9 was. qPCR or western blot data is needed to show that it was not able to lower the mHTT protein (which would be unexpected given it lowered mHTT with a 170 CAG repeat in Figure 2).

Response: As mentioned above, we analyzed *mHTT* using qPCR in the brain of R6/2 mice.

When the mRNA levels of *mHTT* was assessed using qRT-PCR, the expression of *mHTT* decreased in mice treated with dCas9-sgRNA and Cas9-sgRNA (13 weeks of age) (Fig. 4a). We also confirmed the expression of *mHTT* gene in the striatum of R6/2 mice 4 weeks after treatment (8 weeks of age). *mHTT* significantly suppressed in both Cas9-sgRNA- and Cas9-sgRNA-treated HD mice compared to control mice (Supplementary Fig. 5b). These results indicated that both dCas9 and Cas9 treatments reduced *mHTT* by sgRNA targeting the CAG repeat region (Results, page 7, lines 200-207).

We also performed TUNEL staining of the striatum of HD mice to detect cell death. We found significantly fewer TUNEL⁺ cells in the striatum of dCas9-sgRNA-treated R6/2 mice compared to Cas9-sgRNA-treated R6/2 mice (Fig. 4d). These results suggest that dCas9-sgRNA treatment can protect striatal neurons against cell death in brains of HD mice, although both dCas9 and Cas9 treatments reduced *mHTT* by sgRNA targeting the CAG repeat region (Results, page 8, lines 228-232).

Taken together, these results suggest that *mHTT* inhibition using CRISPRi may be a relatively safe treatment than CRISPR-Cas9 nuclease in that it can reduce the cell death

caused by DNA DSBs, consequently delaying functional deterioration (Discussion, page 11, lines 329-332).

7. Pg 5, “This result demonstrates that CRISPRi or CRISPR-Cas9 system has no effect on wild-type mice and the CRISPRi treatment has disease-specific effects.” The findings showed only no observable side-effects. It is possible, and perhaps likely given that a CAG repeat was targeted, that Cas9 induced molecular side-effects.

Response: As you suggested, we investigated molecular side-effects in the brain of wild-type mice treated with CRISPR-Cas9 and CRISPR-dCas9 (Results, page 7, lines 185-196).

To investigate Cas9-induced molecular side-effects, qRT-PCR and immunohistochemistry were performed in the striatum of wild-type mice at 8 weeks of age. *SpCas9* was significantly expressed in both Cas9-sgRNA- and dCas9-sgRNA-treated groups but not in the control group. In addition, *Htt* levels expression was significantly reduced by both Cas9-sgRNA and dCas9-sgRNA compared with Cas9 alone. On the other hand, *Atn3* was significantly lower in Cas9-sgRNA-treated wild-type mice than dCas9-sgRNA- and Cas9 only-treated mice (Supplementary Fig. 4a). When immunohistochemistry was also performed in the striatum of wild-type mice, the densities of β -tubulin⁺ neurons were significantly higher in dCas9-sgRNA-treated mice versus Cas9-sgRNA-treated and control mice, and the densities of β -tubulin⁺ neurons decreased in Cas9-sgRNA-treated mice versus dCas9-sgRNA- and Cas9 only-treated mice (Supplementary Fig. 4b). These results suggest that Cas9-sgRNA treatment has greater molecular side-effects than dCas9-sgRNA treatment.

8. There is no convincing data demonstrating mHTT knockdown in R6/2 mice (a mHTT to DAPI ratio at end-stage is not convincing). Without this, it is not possible to link the improvements in deficits to a decrease in mHTT. This is a major limitation of the study.

Response: As mentioned above, we added results of RT-qPCR analysis in *mHTT* (human specific-mutant HTT) in the striatum of mouse brain at 8 weeks and 13 weeks of age.

When the mRNA levels of *mHTT* was assessed using qRT-PCR, the expression of *mHTT* decreased in mice treated with dCas9-sgRNA and Cas9-sgRNA (13 weeks of age) (Fig. 4a). We also confirmed the expression of *mHTT* gene in the striatum of R6/2 mice 4 weeks after treatment (8 weeks of age). *mHTT* significantly suppressed in both Cas9-sgRNA- and Cas9-sgRNA-treated HD mice compared to control mice (Supplementary Fig. 5b). These results indicated that both dCas9 and Cas9 treatments reduced *mHTT* by sgRNA targeting the CAG repeat region (Results, page 7, lines 200-207).

We also performed TUNEL staining of the striatum of HD mice to detect cell death. We found significantly fewer TUNEL⁺ cells in the striatum of dCas9-sgRNA-treated R6/2 mice compared to Cas9-sgRNA-treated R6/2 mice (Fig. 4d). These results suggest that dCas9-sgRNA treatment can protect striatal neurons against cell death in brains of HD mice, although both dCas9 and Cas9 treatments reduced *mHTT* by sgRNA targeting the CAG repeat region (Results, page 8, lines 228-232).

Taken together, these results suggest that *mHTT* inhibition using CRISPRi may be a relatively safe treatment than CRISPR-Cas9 nuclease in that it can reduce the cell death caused by DNA DSBs, consequently delaying functional deterioration (Discussion, page 11, lines 329-332).

9. Additionally, there is no measurement for off-target effects from the injected tissue. The authors approach involves targeting a CAG repeat and thus has strong potential to influence the expression of other genes.

Response: As you suggested, we performed additional experiments of mRNA-seq and Digenome-seq analysis in the brain of mice.

We confirmed that off-target effects were not observed in mRNA-seq results of the striata of dCas9-sgRNA-treated mice relative to that of Cas9 only-treated mice (Table S3). When we next performed Digenome-seq in the striata of mice treated with Cas9-sgRNA targeting CAG repeat region, the number of *in vitro* cleavage sites was 5152, indicating many off-target sites (Supplementary Fig. 5a) (Results, page 8, lines 238-244).

Minor comments:

1. The statement, “RNAi has some technical limitations related to the volume of solution required for therapy and the delivery method.” The authors should elaborate on what they mean by this. Additionally, the same limitation would very well apply to the CRISPRi approach described by the authors, as both RNAi and Cas9 are being investigated for delivery by similar AAV vectors.

2. “RNAi has some technical limitations related to the volume of solution required for therapy and the delivery method, whereas ASOs reduce wtHTT as well as mHTT expression. Moreover, due to RNA undergoing RNase-mediated degradation, these approaches require repeated treatments for long-term effects. In contrast, genome editing using just a single treatment of CRISPR-Cas9 can lead to permanent repair of specific genes.”

Two thoughts related to these two sentences. One, it implies RNAi requires repeated delivery, which is not true, as these antisense constructs can be delivered by a viral vector. And two, it brings to the readers attention permanent repair by Cas9, which is not the focus of the manuscript. I encourage the authors to be more precise in their language in the lead-up language to their approach and their results.

Response: We greatly appreciate this careful review. As you pointed out, previous studies have shown that RNAi is permanently expressed via delivered by a viral vector. Therefore, we deleted the sentence and revised manuscript in the Introduction as follows:

In the recent search for a cure for HD, several approaches for selectively inhibiting mutant allele expression have been studied, including gene-silencing strategies such as RNA interference (RNAi), antisense oligonucleotides (ASOs), and genome editing tools. Allele-specific suppression methods such as RNAi to selectively inhibit mutant alleles without silencing the *wtHTT* gene have been attempted^{4,6,7}. ASOs are undergoing clinical trials based on positive results derived in mouse and primate models^{8,9}. However, the reduction of *mHTT* using ASOs is not sufficient to provide disease-modifying therapy in the clinical trials of Roche (RG6042) and Wave Life Sciences (WVE-12010 and WVE-120102)¹⁰. In contrast, genome editing system directly targeting the DNA can lead to the permanent repair of specific genes with the potential for greater efficacy.⁸ (Introduction, page 3, lines 51-60).

The latest studies have shown that DNA double-strand breaks (DSBs) induced by CRISPR-Cas9 lead to p53-mediated cell cycle arrest and cell death²¹⁻²³. Moreover, Cas9-induced DNA DSBs can cause mutagenesis by indels at the target site and unintentional off-target sites. On the other hand, CRISPR interference (CRISPRi) system based on catalytically inactive dead Cas9 (dCas9), which does not induce DNA DSBs, allows specific repression of gene expression by sgRNA binding to target genomic loci. It represents an alternative approach to address potential challenges in genetic disorders including inherited retinitis pigmentosa.^{24,25} (Introduction, pages 3-4, lines 74-80).

Reviewer #2 (Remarks to the Author):

Major comments:

1. Abstract: “The CRISPRi resulted in the reduced expression of mHTT with relative preservation of the wild-type HTT in human HD fibroblasts and HD mice” – this sentence should be corrected to precisely describe the results of the study; the preservation of the wt HTT in fibroblasts was not proven by the data provided; HTT expression has not been studied in HD mice (WB or RT-qPCR).

Response: We appreciate your careful review. As you pointed out, we precisely revised the sentence according to the results of this study as follows:

“CRISPRi resulted in the reduced expression of mHTT with relative preservation of the wild-type HTT in human HD fibroblasts. Although both dCas9 and Cas9 treatments reduced mHTT by sgRNA targeting the CAG repeat region, CRISPRi delayed behavioral deterioration and protected striatal neurons against cell death in HD mice.” (Abstract, page 2, lines 33-37).

Expression of wtHTT and mHTT proteins in HD fibroblasts treated with Cas9-sgRNA or dCas9-sgRNA was measured using an antibody (clone name, EPR5526) to distinguish wtHTT and mHTT. Expression of both wtHTT and mHTT was reduced in cells treated with Cas9 or dCas9 compared to the Cas9 only group. In addition, Cas9-treated cells showed a greater reduction in both wtHTT and mHTT expressions compared to Cas9-treated cells in both GM09197 and GM21756 cell lines (Fig. 2d, e). The ratio of wtHTT to mHTT was higher in cells treated with dCas9-sgRNA than in the control, whereas there were no differences between the Cas9-sgRNA and control groups in both GM09197 and GM21756 cell lines (Fig. 2d, e) (Results, page 5, lines 136-143).

Because there are more CAG repeats in the mutant allele, it suggests that dCas9-sgRNA targets the mutant allele more than the normal allele as more CAG repeats would lead to more dCas9 binding, thereby preserving expression of wtHTT compared to Cas9-sgRNA in HD fibroblasts (GM09197, GM21756) (Results, page 6, lines 159-161).

As you recommended, we added results of RT-qPCR analysis in *mHTT* (human specific-mutant HTT) in the striatum of mouse brain at 8 weeks and 13 weeks of age.

When the mRNA levels of *mHTT* was assessed using qRT-PCR, the expression of *mHTT* decreased in mice treated with dCas9-sgRNA and Cas9-sgRNA (13 weeks of age) (Fig. 4a). We also confirmed the expression of *mHTT* gene in the striatum of R6/2 mice 4 weeks after treatment (8 weeks of age). *mHTT* significantly suppressed in both Cas9-sgRNA- and Cas9-sgRNA-treated HD mice compared to control mice (Supplementary Fig. 5b). These results indicated that both dCas9 and Cas9 treatments reduced *mHTT* by sgRNA targeting the CAG

repeat region (Results, page 7, lines 200-207).

2.Introduction: Allele-selective approaches (ASO, RNAi) targeting CAG repeats should be described;

Response: As you suggested, we have now cited references regarding allele-selective approaches (ASO, RNAi) targeting CAG repeats and revised the sentence as follows:

In the recent search for a cure for HD, several approaches for selectively inhibiting mutant allele expression have been studied, including gene-silencing strategies such as RNA interference (RNAi), antisense oligonucleotides (ASOs), and genome editing tools. Allele-specific suppression methods such as RNAi to selectively inhibit mutant alleles without silencing the *wtHTT* gene have been attempted^{4,6,7}. ASOs are undergoing clinical trials based on positive results derived in mouse and primate models^{8,9}. However, the reduction of *mHTT* using ASOs is not sufficient to provide disease-modifying therapy in the clinical trials of Roche (RG6042) and Wave Life Sciences (WVE-12010 and WVE-120102)¹⁰ (Introduction, page 3, lines 51-58).

3.“Delivery of dCas9 with sgRNA targeting the CAG repeat region does not damage the targeted DNA” – unclear, please explain why the authors decided to confirm the known fact that dCas9 does not cut DNA? Why did the authors not analyze the instability of the endogenous CAG tract, since the gRNA targets the CAG repeats?

Response: To determine if the number of CAG repeats is specifically reduced by treatment with Cas9-sgRNA, Cas9-sgRNA or dCas9-sgRNA, we co-transfected with the plasmid containing target sites complementary to the sgRNA into HEK293T cells. Three days after transfection, the target CAG repeat region was deep sequenced and observed that treatment with Cas9-sgRNA, but not dCas9-sgRNA, reduced the number of repeats through cleavage by three to eleven (Fig. 1d, e and Supplementary Fig. 1). Therefore, dCas9 with sgRNA targeting the CAG repeat region circumvents the risk of altering the DNA sequence, a side effect of administering Cas9 nuclease as complicated by the fact that other CAG repeats in the genome would also be targeted (Results, page 4, lines 101-109).

Additionally, we searched the potential off-target sites via Cas9-OFFinder algorithm. When we identified mRNA expression of sorted genes, off-target effects in *ABHD1*, *ATN1*, *ATXN3*, *DCP1B*, *KMT2D*, *FOXJ2*, *FZD1M*, *PLEC*, *SATB1*, *ZNF384*, *ZNF395*, and *ZNF853* were not observed in GM09197 fibroblasts treated with dCas9-sgRNA (Fig. 2a). When off-target effects were also observed using Digenome-seq in GM09197 fibroblasts treated with Cas9-sgRNA,

the number of *in vitro* cleavage sites were 648, indicating relatively many off-target sites (Supplementary Fig. 3a). These results indicated that CRISPR-Cas9 is expected to cause DNA damage due to off-target sites and dCas9 treatment without DSBs will be relatively safe (Results, page 5, lines 128-135).

We also confirmed that off-target effects were not observed in mRNA-seq results of the striata of dCas9-sgRNA-treated mice relative to that of Cas9 only-treated mice (Table S3). When we next performed Digenome-seq in the striata of mice treated with Cas9-sgRNA targeting CAG repeat region, the number of *in vitro* cleavage sites was 5152, indicating many off-target sites (Supplementary Fig. 5a) (Results, page 8, lines 233-237).

4. "CRISPRi targeting the CAG repeat region decreases mHTT expression while preserving wtHTT expression". The main problem with these results is the poor quality of western blots, which in my opinion does not allow to measure the wtHTT/mutHTT ratio (see Fig. 2e, and above all Fig. 2f for which only total HTT level should be determined). In addition, it would be very interesting to know how the number of repeats affects the inhibition.

I am not sure if the proof is convincing enough to talk about some preferences in inhibition. Many studies have shown that it is possible to separate mut and wt HTT proteins from HD fibroblasts, even with a smaller difference between the number of glutamins (e.g., publications from D. Corey's lab). Therefore, better quality WB should be demonstrated. Additional WB analysis on normal fibroblast may confirm wt allele preservation. Please provide all the Wblots in the supplementary figures.

Response: According to your comments, we provided the better quality of western blots (Fig. 1e and f). To confirm difference in HTT expression according to the number of CAG repeat, we also performed additional experiments in human HD fibroblasts with 40-50 CAG repeats and human control fibroblasts.

To investigate the effects of this CRISPRi system in HD cells, we used six human fibroblasts: normal fibroblast (GM04775, 24 and 17 CAG repeats in normal allele; GM07492, 21 and 18 CAG repeats in normal allele), HD patient fibroblasts containing 40~50 CAG repeats (GM04022 in which the *mHTT* allele contains 44 CAG repeats versus the 18 repeats in the normal allele; GM04855 in which the *mHTT* allele contains 48 CAG repeats versus the 20 repeats in the normal allele), GM09197 in which the *mHTT* allele contains 180 CAG repeats (versus the 21 repeats in the normal allele), and GM21756 in which the *mHTT* allele contains 70 CAG repeats (versus the 15 repeats in the normal allele). GM09197 showed a clear difference between the sizes of the normal and mutant proteins whereas GM21756 showed

similar sizes between the normal and mutant proteins on Western blots (Supplementary Fig. 2a) (Results, page 5, lines 113-123).

In normal fibroblasts, expression of HTT protein showed a tendency to decrease in the Cas9-sgRNA-treated cells when an EPR5526 antibody was used to detect HTT protein (Supplementary Fig. 3f). The HTT protein expression using 4C8 antibody was ultimately reduced in Cas9- and dCas9-treated groups while its counterpart Cas9 only group did not show any reduction (Supplementary Fig. 3g). Moreover, Cas9-sgRNA and dCas9-sgRNA reduced the HTT protein in HD fibroblasts containing 40-50 CAG (Supplementary Fig. 3j, k). These results showed that the mentioned strategies using sgRNA targeting CAG region can reduce mHTT and wtHTT in both normal fibroblasts and HD fibroblasts.

Because there are more CAG repeats in the mutant allele, it suggests that dCas9-sgRNA targets the mutant allele more than the normal allele as more CAG repeats would lead to more dCas9 binding, thereby preserving expression of wtHTT compared to Cas9-sgRNA in HD fibroblasts (GM09197, GM21756) (Results, page 6, lines 150-161).

5.Line 95, please explain the off-target analysis in more detail; how were the candidates selected? How long are the CAG repeats in these genes? The list does not include e.g., ATXN3 gene, which contains ~23CAG in normal population. Additional evidence is needed to support the conclusion that the presented strategy does not generate off-target effects.

Response: We searched the potential off-target sites via Cas9-OFFinder algorithm. We identified the mRNA expression of sorted genes including *ATXN3* to confirm off-target effect in HD fibroblast treated with dCas9-sgRNA. Digenome-seq analysis was also performed in HD fibroblast treated with Cas9-sgRNA.

When we identified mRNA expression of sorted genes, off-target effects in *ABHD1*, *ATN1*, *ATXN3*, *DCP1B*, *KMT2D*, *FOXJ2*, *FZD1M*, *PLEC*, *SATB1*, *ZNF384*, *ZNF395*, and *ZNF853* were not observed in GM09197 fibroblasts treated with dCas9-sgRNA (Fig. 2a). When off-target effects were also observed using Digenome-seq in GM09197 fibroblasts treated with Cas9-sgRNA, the number of *in vitro* cleavage sites were 648, indicating relatively many off-target sites (Supplementary Fig. 3a). These results indicated that CRISPR-Cas9 is expected to cause DNA damage due to off-target sites and dCas9 treatment without DSBs will be relatively safe (Results, page 5, lines 128-135).

6.It is not clear why, in the murine model, HTT protein and / or transcript levels were not directly tested to confirm the molecular effects.

Response: Based on your comment, we added results of RT-qPCR analysis in *mHTT* (human specific-mutant HTT) in the striatum of mouse brain at 8 weeks and 13 weeks of age.

When the mRNA levels of *mHTT* was assessed using qRT-PCR, the expression of *mHTT* decreased in mice treated with dCas9-sgRNA and Cas9-sgRNA (13 weeks of age) (Fig. 4a). We also confirmed the expression of *mHTT* gene in the striatum of R6/2 mice 4 weeks after treatment (8 weeks of age). *mHTT* significantly suppressed in both Cas9-sgRNA- and Cas9-sgRNA-treated HD mice compared to control mice (Supplementary Fig. 5b). These results indicated that both dCas9 and Cas9 treatments reduced *mHTT* by sgRNA targeting the CAG repeat region (Results, page 7, lines 200-207).

7.Lines 214, 215 – although in the graph presented in Fig. 4c we can see significantly (*P<0.05) lower level of HTT compared to the control, it should be kept in mind that the staining methods should not be used alone for evaluation of gene expression, but as accompanying more quantitative methods.

Response: As above mentioned, we added results of RT-qPCR analysis in *mHTT* (human specific-mutant HTT) in the striatum of mouse brain at 8 weeks and 13 weeks of age (Results, page 7, lines 200-207).

8.Please discuss the possible mechanisms underlying the observed effects.

Response: As you suggested, we provided the possible mechanisms in this experiment.

We found that treatment of dCas9 with sgRNA targeting CAG repeats suppressed mHTT relative to wtHTT in human fibroblasts derived from HD patient because there are more CAG repeats in the mutant allele. These results indicated that the sgRNA is efficient in targeting the CAG repeat region and that dCas9-sgRNA preserves expression wtHTT more than mtHTT compared to Cas9-sgRNA (Discussion, page 9, lines 288-292)

Importantly, in the striatum of dCas9-sgRNA-treated mice, the expression of *mHTT* was significantly reduced compared to Cas9 only-treated mice, and TUNEL⁺ cells were significantly reduced compared to Cas9-sgRNA-treated mice. Taken together, these results suggest that *mHTT* inhibition using CRISPRi may be a safer treatment than CRISPR-Cas9 nuclease in that it can reduce the cell death caused by DNA DSBs, consequently delaying functional deterioration (Discussion, pages 10-11, lines 327-332).

9. Generally there are many sentences with incorrect references, e.g., line 21 (ref. 1,2); line 23 (ref. 4,5); lines 36, 37, 39,... I suggest to cite the original works and not the reviews (e.g. line 37). Please check all the citations in the text.

Response: As you suggested, we have edited the citations in the manuscript.

Minor points:

1. Please add information about virus MOI used in cell experiments
2. Line 114, indicate the CAG tract length in the R6/2 mouse model
3. Line 188, to be more specific - CAG tract rather than polyQ (PROTEIN)
4. Line 274, 314, virus MOI is not specified
5. Line 319, CAG repeat number should be specified
6. Some important references are missing e.g., those concerning allele-selective CAG targeting by ASO/RNAi; different CRISPR-Cas9-based approaches toward HD,
7. Fig. 2b, legend, line 529 – the legend that explains this experiment is missing
8. It is not clear what is the difference between Fig. 2c and d.
9. Fig. S2b – please specify the cell line

Response: Based on your comments, we have revised the points in the manuscript.

Reviewer #3 (Remarks to the Author):

This publication looks to investigate the use of CRISPRi as a treatment for HD in human cell models and HD mouse models. They assessed behavioural phenotypes and neuronal survival in dCas9-treated mice. Motor function was assessed using open field test and rotarod, both showing an improvement with the dCas9-gRNA- though motor decline still occurred at a substantial rate. These findings suggest that CRISPRi may be used to delay disease onset or progression, which would be valuable in the context of human disease.

I think that this manuscript is important for the future utility of CRISPRi or genome-editing in the context of treatment for disease. The manuscript may benefit from a thorough explanation of experimental design and strategy and a more well-rounded discussion, discussing the translation of CRISPRi as a potential therapeutic treatment in humans.

Response: We greatly appreciate your positive statement.

My major criticism is the use of the R6/2 mouse model which is transgenic for expanded human exon 1 of mHTT only, I think this methodology only really shows its translational benefit if also tested in a full-length knock in model such as the Q111 or zQ175 mouse or even the YAC 128 or BACHD mice. Those are the mouse models the field really focuses on now for therapeutic translation potential. This paper did not convince me of CRISPRi as a potential novel therapeutic for Huntington's disease.

Response: As you pointed out, we added the limitation of the HD mouse model in the Discussion as follows (Page 10, lines 312-319):

The R6/2 mouse model containing approximately 160 CAG repeats is one of the most widely used in HD studies because it displays typical disease phenotypes with early onset, severe motor decline and short lifespan.^{1,36} It can be used to study novel treatment with the advantage of being able to easily confirm therapeutic effect because the disease phenotype appears obvious. However, because the R6/2 model is truncated N-terminus fragment models, it has limitations in studying full-length protein function. In the future, therapeutic effects should be examined in the HD mouse model with a full-length mHTT such as Q111, zQ175, YAC128 or BACHD which exhibit similar neuropathology of the human with HD³¹.

Other points:

Introduction:

1. I think it would be useful for the reader in the introduction if the authors explained how CRISPRi works. The introduction describes other techniques but does not provide much insight on the reasoning behind CRISPRi or the proposed mechanism of action. Perhaps including references to other diseases where CRISPRi has been utilised or proved effective. There is reference to the desired knockdown of mHTT only, though it's not clear how this is selectively achieved with CRISPRi.

Response: As you recommended, we have provided the insight on the CRISPRi and the proposed mechanism of action in the Introduction as follows (Pages 3-4, lines 74-86):

The latest studies have shown that DNA double-strand breaks (DSBs) induced by CRISPR-Cas9 lead to p53-mediated cell cycle arrest and cell death²¹⁻²³. Moreover, Cas9-induced DNA DSBs can cause mutagenesis by indels at the target site and unintentional off-target sites. On the other hand, CRISPR interference (CRISPRi) system based on catalytically inactive dead Cas9 (dCas9), which does not induce DNA DSBs, allows specific repression of gene expression by sgRNA binding to target genomic loci. It represents an alternative approach to address potential challenges in genetic disorders including inherited retinitis pigmentosa.^{24,25}

This study investigated the therapeutic effects of dCas9 with sgRNA targeting the CAG repeat region in human HD fibroblasts and HD transgenic mice. We asked that delivery of dCas9-sgRNA suppresses *mHTT* in human HD fibroblasts, and the dCas9 treatment can mitigate behavioral deterioration in a mouse model of HD. Here, it has been shown that DNA DSB-free CRISPRi is a potential therapy for HD that can compensate for the shortcoming of CRISPR-Cas9 nuclease in reducing cell death and delaying disease progression.

2. There are a number of sentence errors in the introduction;

- line 27; 'there has'
- lines 27-29; slight rewording perhaps
- line 45; doesn't read well, perhaps remove the word 'that'
- sections could be written to flow slightly better, the penultimate paragraph feels like a collection of statements rather than a congruous section.

Response: Based on your comments, we revised sentence errors in the Introduction.

3. Line 39 mentions ASOs though it should be noted that there are ASOs that reduce both wtHTT/mHTT (Tomkinson) and allele-selective ASOs that have been used in clinical trials (Wave).

Response: As you suggested, we have now noted the sentence regarding clinical trials of allele-selective approaches ASO in the Introduction as follows (Page 3, lines 51-60):

In the recent search for a cure for HD, several approaches for selectively inhibiting mutant allele expression have been studied, including gene-silencing strategies such as RNA interference (RNAi), antisense oligonucleotides (ASOs), and genome editing tools. Allele-specific suppression methods such as RNAi to selectively inhibit mutant alleles without silencing the *wtHTT* gene have been attempted^{4,6,7}. ASOs are undergoing clinical trials based on positive results derived in mouse and primate models^{8,9}. However, the reduction of *mHTT* using ASOs is not sufficient to provide disease-modifying therapy in the clinical trials of Roche (RG6042) and Wave Life Sciences (WVE-12010 and WVE-120102)¹⁰.

Results:

4. Figure 1: I am not convinced that Figure 1a schematic accurately explains the system design. It perhaps needs an accompanying explanation that this is due to more Cas9 complexes binding to the elongated polyQ stretch- if this is the proposed way in which it preferentially reduces mHTT.

Figure 1 is titled 'The sgRNA with CAG as PAM sequence reduces the number of CAG repeats'. With what we know about repeat instability and reducing the number of CAG repeats being seen as beneficial, this could perhaps be reworded to indicate that in this instance- this was not the desired effect. The manuscript stance focuses on the efficacy of dCas9, so this should be the focus of the figure- rather than what the Cas9-sgRNA does. "dCas9-sgRNA aims to reduce mHTT expression without altering CAG repeat length"

Response: As you recommended, we have additionally explained in the legend of Figure 1.

"A CRISPRi was designed to bind to the polyglutamine tract and aims to reduce mHTT expression without altering CAG repeat length. Relative repression between the mutant allele and normal allele is caused by the different number of CAG repeats. Cas9 complexes bind to the mutant allele more than the normal allele due to the elongated polyglutamine stretch."

5. Following on from point 4- provide more detail in the results section as to how the sgRNA preferentially targets the mHTT allele rather than the wtHTT allele.

Response: As you suggested, we provided the description in the Results as follows:

Because there are more CAG repeats in the mutant allele, it suggests that dCas9-sgRNA targets the mutant allele more than the normal allele as more CAG repeats would lead to more dCas9 binding, thereby preserving expression of wtHTT compared to Cas9-sgRNA in HD fibroblasts (GM09197, GM21756) (Page 6, lines 158-161).

6. Figure 2: Shows Cas9 expression in fibroblasts 10 days after treatment. Shows HTT KD in two fibroblast lines. Compares KD ratios in HTT and mHTT. I think they interpret this data correctly, though would be good to see full blots in supplementary.

Response: According to your comments, we provided the better quality of western blots (Fig. 1e and f). To confirm difference in HTT expression according to the number of CAG repeat, we also performed additional experiments in human HD fibroblasts with 40-50 CAG repeats and human control fibroblasts.

To investigate the effects of this CRISPRi system in HD cells, we used six human fibroblasts: normal fibroblast (GM04775, 24 and 17 CAG repeats in normal allele; GM07492, 21 and 18 CAG repeats in normal allele), HD patient fibroblasts containing 40~50 CAG repeats (GM04022 in which the *mHTT* allele contains 44 CAG repeats versus the 18 repeats in the normal allele; GM04855 in which the *mHTT* allele contains 48 CAG repeats versus the 20 repeats in the normal allele), GM09197 in which the *mHTT* allele contains 180 CAG repeats (versus the 21 repeats in the normal allele), and GM21756 in which the *mHTT* allele contains 70 CAG repeats (versus the 15 repeats in the normal allele). GM09197 showed a clear difference between the sizes of the normal and mutant proteins whereas GM21756 showed similar sizes between the normal and mutant proteins on Western blots (Supplementary Fig. 2a) (Results, page 5, lines 113-123).

In normal fibroblasts, expression of HTT protein showed a tendency to decrease in the Cas9-sgRNA-treated cells when an EPR5526 antibody was used to detect HTT protein (Supplementary Fig. 3f). The HTT protein expression using 4C8 antibody was ultimately reduced in Cas9- and dCas9-treated groups while its counterpart Cas9 only group did not show any reduction (Supplementary Fig. 3g). Moreover, Cas9-sgRNA and dCas9-sgRNA reduced the HTT protein in HD fibroblasts containing 40-50 CAG (Supplementary Fig. 3j, k). These results showed that the mentioned strategies using sgRNA targeting CAG region can reduce mHTT and wtHTT in both normal fibroblasts and HD fibroblasts (Results, page 6, lines 150-167).

7. Line 102-104: In addition, dCas9-treated cells showed that wtHTT expression significantly increased compared to Cas9-treated cells in both GM09197 and GM21756 cell lines (Fig. 2e, f)- this sentence is not correct, wtHTT was still reduced in dCas9-treated cells, Cas9-treated cells just showed a greater reduction in wtHTT and mHTT expression.

Response: We agree with your comments. We revised the sentence as follows:

In addition, Cas9-treated cells showed a greater reduction in both wtHTT and mHTT expressions compared to Cas9-treated cells in both GM09197 and GM21756 cell lines (Fig. 2d, e) (Results, page 5, lines 139-140).

8. Lines 108-110 suggest that findings from supplementary figure 2b indicate greater knockdown of mHTT than wtHTT. Supplementary figure 2b does indeed show reduction of mHTT in dCas9-sgRNA treated cells, however the use of antibody 1C2 which only probes mHTT means that we cannot infer these findings from supplementary figure 2b. This statement should be made in reference to Figure 2e and f, where antibodies probing both alleles have been used.

Response: As you pointed out, we revised the sentence as follows:

When total HTT expression was measured using 4C8 antibody in both GM09197 and GM21756 cell lines, total HTT was significantly reduced by Cas9-sgRNA and dCas9-sgRNA treatment (Supplementary Fig. 2c, d). Following the Cas9-sgRNA or dCas9-sgRNA treatment, we observed a significant reduction in mHTT protein levels when an mHTT antibody (clone name, 1C2) was used to detect polyglutamine region in a GM0919 cell line (Supplementary Fig. 2e, f) (Results, pages 5-6, lines 144-149).

9. Figure 3: The authors show that dCas9-treated animals perform better on the rotarod at various time points up to 13 weeks. They also show increased exploration in dCas9-treated animals in the open field test. I think the author should make clear when they say 'ameliorate' whether they mean a delay in decline or a change in the rate of decline. One would suggest that treatment delays initial motor symptoms, whereas the other would suggest a consistent benefit to phenotype throughout the mouse's life. Whilst there is an improvement of motor function on the rotarod, animals in all groups show severe motor decline- so I do not think you can report the amelioration of disease progression, alleviation perhaps. It would be good to include a brief description of R6/2 model of HD and its severity.

Response: The meaning of our data indicated that treatment delays motor deterioration.

As you recommended, we have changed the word “amelioration or alleviation” to “delay” throughout the manuscript including title.

We also included a brief description of R6/2 model of HD and its severity as follows:

The R6/2 mouse model containing approximately 160 CAG repeats is one of the most widely used in HD studies because it displays typical disease phenotypes with early onset, severe motor decline and short lifespan. It can be used to study novel treatment with the advantage of being able to easily confirm therapeutic effect because the disease phenotype appears obvious (Discussion, page 10, lines 312-316).

Reviewers' comments:

Reviewer #1 (Remarks to the Author):

This revised manuscript addresses several key concerns raised in my initial review, chief among them data demonstrating mHTT lowering in the striatum of injected mice.

However, several concerns remain. For one, I have questions about the in vitro data demonstrating whether dCas9 can selectively lower mHTT versus wtHTT. And two, the manuscript remains confusing and difficult to interpret. It is highly recommended that the authors edit their manuscript for maximum clarity.

Comments:

1. The statement, "The reduction of mHTT using ASOs is not sufficient to provide disease-modifying therapy in the clinical trials of Roche (RG6042) and Wave Life Sciences (WVE-12010 and WVE-120102)" is not fully accurate. The Roche trial was halted due to a poor risk/benefit profile, while the Wave trial showed no target engagement. The authors should update this statement.

2. The meaning of the statement in line 72 is not fully clear, particularly in the context of the next sentence. The authors should clarify their language

3. Line 82, should be, "We asked if delivery..."

4. Lines 99-102, this sentence is confusing. The authors appear to be describing their reporter system but in the context of their results?

5. Line 106, do the authors mean they reduced the number of repeats from 11 to 3?

6. The organization of the results on Pg. 5 are difficult to follow. I recommend discussing the targeting results after the mHTT/wtHTT ratio determinations and BEFORE the description of the OT studies.

7. Regarding the newly added studies for measuring off-target effects, more information must be provided on how the sites were identified by Cas OFFinder. What was the cut-off, how many sites were found, etc. The same applies to the diGenome-seq experiments. What was the "strength" of the scores for the OT this? It is difficult to interpret the findings as is.

8. For Figure 2, the author aim to selectively reduce mHTT, but is there a significant difference in mHTT protein versus wtHTT protein for the dCas9 targeting group? This analysis is missing. The authors are relying on a ratio to make their point between the groups, but I do not think this answers the whole question.

Further, the authors state, "Because there are more CAG repeats in the mutant allele, it suggests that dCas9-sgRNA targets the mutant allele more than the normal allele as more CAG repeats would lead to more dCas9 binding, thereby preserving expression of wtHTT compared to Cas9-sgRNA in HD fibroblasts (GM09197, GM21756)." I had a difficult time finding the data indicating this.

9. Line 164, did the authors inject LV vector encoding Cas9? They should state the delivery system in the main text. Additionally, which promoter was used to drive Cas9 and was IHC done to identify the cells that expressed it?

10. The final paragraph of the results repeats several of the results (mHTT silencing for example)

described earlier in the section.

11. The authors interestingly was a fixed rotated speed versus an accelerating speed. Is data available on the groups performance on an accelerating rotarod?

Reviewer #2 (Remarks to the Author):

I appreciate the authors' efforts to improve the manuscript. However, many issues still need to be clarified.

1. Graphical abstract: do the authors propose lentiviral vectors as therapeutic approach for patients? Lentiviruses integrate into the genome so other vectors are rather proposed now (e.g. AAV). I suggest to remove the human figure from the abstract.

2. Introduction: I still have doubts about the selection of the works cited:

e.g., "Huntington's disease (HD) is a progressive neurodegenerative disease caused by a CAG repeat expansion at the exon 1 location of the huntingtin (HTT) gene (1-3)".

Ref.1 Cowin, R. M. et al. Onset and progression of behavioral and molecular phenotypes in a novel congenic R6/2 line exhibiting intergenerational CAG repeat stability. PLoS One (2011).

Ref.2 Scheuing, L., et al., Preclinical and clinical investigations of mood stabilizers for Huntington's disease: what have we learned? Int J Biol Sci 10, (2014).

e.g., "HD accompanied by impairments of motor function, cognition, and psychiatric symptoms can often lead to death (4,5)".

Ref.4 Liu, et al., MicroRNA-124 slows down the progression of Huntington's disease by promoting neurogenesis in the striatum. Neural Regen Res 10, (2015).

Ref.5 Johnson, R., et al., microRNA-based gene dysregulation pathway in Huntington's disease. Neurobiol Dis (2008)

.....and in many other sites.

3. Lines 56-60, "However, the reduction of mHTT using ASOs is not sufficient to provide disease-modifying therapy in the clinical trials of Roche (RG6042) and Wave Life Sciences (WVE-12010 and WVE-120102)". I do not agree with the authors to describe the results of these two different studies in the same way. In fact, WVE molecules did not induce effective silencing of mHTT in clinical trials.

4. Line 65 – please add „mouse“ model

5. The authors still have not described the mechanism: is it an allele-preferential transcriptional repression like in Zeitler et al., ? There is no information that typical CRISPRi works mainly in the promoter regions. Does the presence of CAG repeats in exon 1 of the HTT gene is important for the mechanism? How long a CAG tract must be for an efficient transcription block?

6. Unlike typical genome editing with wtCas9, the proposed strategy requires constant expression of dCas9 and the effect is not permanent - it should be clearly stated.

7. Considering the high level of wtHTT suppression, the phrase "preservation of wtHTT expression" is imprecise

Reviewer #3 (Remarks to the Author):

My comments are now answered with the data, and I feel they have reworded sections which are better interpretations of their data. However, I do not feel that the manuscript reads clearly throughout.

Minor comments:

1. Background discussion of other HTT lowering techniques is vague and not completely clear what the findings were. Would benefit from more detail.

Allele-selective ASOs reduce mHTT in CSF but may not have had an effect on behaviour due to?

2. Lines 69-73 feel a bit out of place, perhaps put them after line 80 so it flows better.

3. Line 189: more context needed. "To assess the effect on other CAG repeat regions we looked at Atxn3 levels.."

4. Lines 292-301: Altered HTT and mHTT expression in WT fibroblasts, but no effect on WT Atxn3 levels? The Cas9 is not specific to HTT, so why is dCas9 lowering WT HTT but not lowering other WT loci? Looking at more loci than just Atxn3 would be advantageous.

5. Line 300: " These results suggest that CRISPRi does not significantly regulate the expression of normal gene with CAG repeats." You do report lowered expression of WT HTT with both systems though.

Response to Reviewers' Comments

Manuscript ID COMMSBIO-22-0171A, entitled "DNA double-strand break-free CRISPR interference alleviates Huntington's disease progression in mice"

Reviewer #1 (Remarks to the Author):

This revised manuscript addresses several key concerns raised in my initial review, chief among them data demonstrating mHTT lowering in the striatum of injected mice.

However, several concerns remain. For one, I have questions about the in vitro data demonstrating whether dCas9 can selectively lower mHTT versus wtHTT. And two, the manuscript remains confusing and difficult to interpret. It is highly recommended that the authors edit their manuscript for maximum clarity.

Comments:

1. The statement, "The reduction of mHTT using ASOs is not sufficient to provide disease-modifying therapy in the clinical trials of Roche (RG6042) and Wave Life Sciences (WVE-12010 and WVE-120102)" is not fully accurate. The Roche trial was halted due to a poor risk/benefit profile, while the Wave trial showed no target engagement. The authors should update this statement.

Response: We greatly appreciate your careful review. We agree with your comments.

As you recommended, we updated the sentence as follows (Introduction, page 3, lines 57-60):

In the clinical trials of ASOs in HD, Roche's phase III drug Tominersen reduced HTT levels but was halted due to a poor risk-benefit profile, while Wave's drugs (WVE-12010 and WVE-120102) did not induce effective silencing of *mHTT* mRNA.

2. The meaning of the statement in line 72 is not fully clear, particularly in the context of the next sentence. The authors should clarify their language

Response: As you pointed out, we revised the sentence as follows (Introduction, page 4, lines 82, 83):

Therefore, we used sgRNA containing CAG PAM sequence to target CAG repeat region in treating HD.

3. Line 82, should be, “We asked if delivery...”

Response: Based on your comments, we have revised the points in the manuscript (Introduction, page 4, line 96).

4. Lines 99-102, this sentence is confusing. The authors appear to be describing their reporter system but in the context of their results?

Response: As you pointed out, we revised the sentence as follows (Results, page 5, lines 112-119):

To test the efficacy of the CAG repeat-targeting sgRNA, Cas9-sgRNA (with CAG PAM) and a reporter plasmid containing the sgRNA target site were co-transfected into HEK293T cells. If Cas9-sgRNA make DSBs in the target site in the reporter, nucleotide insertions or deletions at the target site can lead to eGFP expression. Our results show that 34% eGFP expressions were observed in HEK293T cells treated with Cas9-sgRNA. These results suggest that Cas9-RNA with CAG repeat-targeted sgRNA significantly acts on target site to make DSBs.

5. Line 106, do the authors mean they reduced the number of repeats from 11 to 3?

Response: As you pointed out, we revised the sentence as follows (Results, page 5, lines 122-124):

The target CAG repeat region was deep sequenced at three days after transfection and it was observed that treatment with Cas9-sgRNA, but not dCas9-sgRNA, reduced the number of repeats through cleavage from 27 to 3-11.

6. The organization of the results on Pg. 5 are difficult to follow. I recommend discussing the targeting results after the mHTT/wtHTT ratio determinations and BEFORE the description of the OT studies.

Response: We agree with your comments. We reorganized the sentence as your comments.

7. Regarding the newly added studies for measuring off-target effects, more information must be provided on how the sites were identified by Cas OFFinder. What was the cut-off, how many sites were found, etc. The same applies to the diGenome-seq experiments. What was the “strength” of the scores for the OT this? It is difficult to interpret the findings as is.

Response: We greatly appreciate your careful review. We described the additional results as follows:

Cas-OFFinder algorithm was then used to analyze potential off-target sites for sgRNA targeting CAG repeat region in HTT gene. We obtained 199 sites for sgRNA targeting CAG repeat region from Cas-OFFinder by allowing for up to 2 bp mismatches and no DNA or RNA bulges (Results, page 5, lines 140-143).

In addition, we confirmed several genes containing CAG repeats associated with polyglutamine-associated diseases, such as *ATXN1*, *ATN1*, *ATXN7*, *CACNA1A*, *TBP*, and on-target gene, *HTT* in both human HD and human control fibroblasts. In human control fibroblasts, after Cas9-sgRNA treatment, the expression of *ATXN1*, *ATN1*, and *HTT* genes significantly decreased compared to the Cas9 only- and dCas9-sgRNA treatment. In HD fibroblast containing 40-50 CAG, *ATN1* and *HTT* were significantly suppressed in the Cas9-sgRNA-treated group compared to the other groups and *HTT* gene expression was decreased in the dCas9-sgRNA group than Cas9 only group. The Cas9-sgRNA and dCas9-sgRNA significantly reduced the *HTT* gene in GM21756 cell line (HD fibroblasts) compared to Cas9 only group, and treatment with Cas9-sgRNA suppressed *HTT* gene expression more than treatment with dCas9-sgRNA. In GM09197 fibroblast, expression of *ATN1* and *TBP* genes was significantly suppressed by the Cas9-sgRNA treatment and *HTT* gene expression was significantly decreased in both the Cas9-sgRNA group and dCas9-sgRNA group compared to Cas9 only group. These results showed that suppression of the potential off-target genes was not observed in the dCas9-sgRNA-treated group and suggested that CRISPR-Cas9 cuts the DNA to shut the targeted gene off, but sgRNA targeting CAG repeat regions may have side effects of mutating normal genes. Consequently, both in-silico and qRT-PCR for our sgRNA were used to identify potential off-targets (Results, page 6, lines 149-166).

Digenome-seq is one method for profiling genome-wide off-target sites of Cas9 nuclease²⁹. Using Digenome-seq, we were able to identify Cas9 system off-target sites that were targeting HTT sites. A high DNA cleavage score indicates that Cas9 nuclease causes double-strand breaks at those locations (Results, page 6, lines 167-170).

8. For Figure 2, the author aim to selectively reduce mHTT, but is there a significant difference in mHTT protein versus wtHTT protein for the dCas9 targeting group? This analysis is missing. The authors are relying on a ratio to make their point between the groups, but I do not think this answers the whole question.

Further, the authors state, “Because there are more CAG repeats in the mutant allele, it suggests that dCas9-sgRNA targets the mutant allele more than the normal allele as more CAG repeats would lead to more dCas9 binding, thereby preserving expression of wtHTT compared to Cas9-sgRNA in HD fibroblasts (GM09197, GM21756).” I had a difficult time finding the data indicating this.

Response: According to your comments, we described the additional results as follows (Results, page 7, lines 195-200):

We provided analysis to difference in mHTT protein versus wtHTT protein for the all groups (Supplementary Fig. 3g,h). Unlike the groups treated with Cas9-sgRNA or Cas9 only, the

mHTT protein was significantly decreased compared to the wtHTT protein in the group treated with dCas9-sgRNA in both GM09197 and GM21756 fibroblasts. This result suggests that dCas9 has more activity in the mutant allele with longer CAG repeats than in the wild-type allele.

9. Line 164, did the authors inject LV vector encoding Cas9? They should state the delivery system in the main text. Additionally, which promoter was used to drive Cas9 and was IHC done to identify the cells that expressed it?

Response: Based on your comments, have revised the points in the manuscript.

We used lentiviral vector containing sgRNA and CRISPR-Cas9. The sgRNA targeting CAG repeat region were cloned into the LentiCRISPR v2 plasmid (plasmid #52961, Addgene). For HTT gene suppression, LentiCRISPR v2 plasmid with U6 promoter driving sgRNA expression and EF-1 alpha promoter driving the expression of Puromycin-T2A-HA-NLS-dCas9-NLS (or Puromycin-T2A-Flag-NLS-Cas9-NLS) was used (Figure 1a) (Methods, page 14, lines 419-421).

To confirm the expression of Cas9 *in vivo*, we performed immunohistochemistry in HD mouse brain-treated with Lentiviral vector containing CRISPR system. SpCas9⁺ cells were expressed by dCas9-sgRNA-, Cas9-sgRNA- or Cas9 only-treatment in R6/2 mice brain at 4 weeks after treatment (Supplementary Fig. 7d) (Results, page 10, lines 308-312).

10. The final paragraph of the results repeats several of the results (mHTT silencing for example) described earlier in the section.

Response: As you pointed out, we have confirmed the repeated results of mHTT silencing, and Supplementary Figure 7 shows the results of the experiment in 8-week-old R6/2 mice at 4 weeks after treatment with Cas9 only, Cas9-sgRNA, and dCas9-sgRNA (Results, page 10, lines 308-312).

11. The authors interestingly was a fixed rotated speed versus an accelerating speed. Is data available on the groups performance on an accelerating rotarod?

Response: As you pointed out, we mentioned about accelerating speed rotarod as follows (Results, page 8, lines 223-229):

Rotarod apparatus is the most used neuro-behavior tests for assessing motor coordination and balance in rodents. Our rotarod results showed that motor function was maintained at constant speed in dCas9-sgRNA-treated Huntington mice (Figure 3b). In addition, although there were no dramatic differences in latency time at accelerating speed, the latency periods in the dCas9-sgRNA group were significantly improved at six to eleven weeks of age compared with the Cas9-sgRNA, Cas9-only and PBS groups (Supplementary Fig. 5). These results suggest that the constant speed rotarod may be more sensitive to motor function than the accelerated speed rotarod in the CRISPR-Cas9 system treated with HD mice, though acceleration speed may yield specific results.

Reviewer #2 (Remarks to the Author):

I appreciate the authors' efforts to improve the manuscript. However, many issues still need to be clarified.

1. Graphical abstract: do the authors propose lentiviral vectors as therapeutic approach for patients? Lentiviruses integrate into the genome so other vectors are rather proposed now (e.g. AAV). I suggest to remove the human figure from the abstract.

Response: We greatly appreciate your careful review. According to your comments, we revised graphic abstract.

2. Introduction: I still have doubts about the selection of the works cited:

e.g., "Huntington's disease (HD) is a progressive neurodegenerative disease caused by a CAG repeat expansion at the exon 1 location of the huntingtin (HTT) gene (1-3)".

Ref.1 Cowin, R. M. et al. Onset and progression of behavioral and molecular phenotypes in a

novel congenic R6/2 line exhibiting intergenerational CAG repeat stability. PLoS One (2011).
Ref.2 Scheuing, L., et al., Preclinical and clinical investigations of mood stabilizers for Huntington's disease: what have we learned? Int J Biol Sci 10, (2014).
e.g., "HD accompanied by impairments of motor function, cognition, and psychiatric symptoms can often lead to death (4,5)".
Ref.4 Liu, et al., MicroRNA-124 slows down the progression of Huntington's disease by promoting neurogenesis in the striatum. Neural Regen Res 10, (2015).
Ref.5 Johnson, R., et al., microRNA-based gene dysregulation pathway in Huntington's disease. Neurobiol Dis (2008)
.....and in many other sites.

Response: As you pointed out, we have corrected the citation in the manuscript (Introduction, page 3, lines 43-45).

3. Lines 56-60, "However, the reduction of mHTT using ASOs is not sufficient to provide disease-modifying therapy in the clinical trials of Roche (RG6042) and Wave Life Sciences (WVE-12010 and WVE-120102)". I do not agree with the authors to describe the results of these two different studies in the same way. In fact, WVE molecules did not induce effective silencing of mHTT in clinical trials.

Response: As you recommended, we updated the sentence as follows (Introduction, page 3, lines 57-60):

In the clinical trials of ASOs in HD, Roche's phase III drug Tominersen reduced HTT levels but was halted due to a poor risk-benefit profile, while Wave's drugs (WVE-12010 and WVE-120102) did not induce effective silencing of *mHTT* mRNA.

4. Line 65 – please add "mouse" model

Response: Based on your comments, we have revised the points in the manuscript (Introduction, page 3, line 66).

5. The authors still have not described the mechanism: is it an allele-preferential transcriptional repression like in Zeitler et al., ? There is no information that typical CRISPRi works mainly in the promoter regions. Does the presence of CAG repeats in exon 1 of the HTT gene is

important for the mechanism? How long a CAG tract must be for an efficient transcription block?

Response: As you recommended, we have described the mechanism of our study (Introduction, page 4, lines 84-90).

Since wtHTT is thought to play an important role in the brain as a therapeutic strategy, we designed a dCas9 and sgRNA complex that binds to the CAG repeat region in *HTT* gene and represses *mHTT* transcription of the mutant allele relative to that of the wild-type allele. CRISPR/Cas9 system recognizes 23bp including the NGG PAM sequence. Theoretically, there should be at least 8 CAG repeats for Cas9 nucleases to bind to (3bp repeat x 8 =24bp). Therefore, our *mHTT* suppression strategy was designed with the expectation that the longer the CAG repeat, the greater the suppression.

6. Unlike typical genome editing with wtCas9, the proposed strategy requires constant expression of dCas9 and the effect is not permanent - it should be clearly stated.

Response: As you recommended, we have described that because of the non-constitutive expression of dCas9, a strategy for constant expression was required (Introduction, page 4, lines 91-94):

The wild-type CRISPR-Cas9 can be sufficient for genome editing in a short duration but CRISPRi strategy using dCas9 requires the sustained expression of dCas9 to inhibit transcription. Therefore, we used a lentiviral vector to constitutively express dCas9 in the HD models because the effect of dCas9 expression is not permanent.

7. Considering the high level of wtHTT suppression, the phrase "preservation of wtHTT expression" is imprecise

Response: As you pointed out, we revised the phrase "CRISPRi strategy relatively preserves the expression of wtHTT and protects striatal neurons against DNA damage in HD" (Concluding remarks, page 13, lines 407-409), and "preservation of wtHTT protein levels" to "relative preservation of wtHTT protein levels" (Legend, page 26, lines 791, 799).

Reviewer #3 (Remarks to the Author):

My comments are now answered with the data, and I feel they have reworded sections which are better interpretations of their data. However, I do not feel that the manuscript reads clearly throughout.

Minor comments:

1. Background discussion of other HTT lowering techniques is vague and not completely clear what the findings were. Would benefit from more detail.

Allele-selective ASOs reduce mHTT in CSF but may not have had an effect on behaviour due to?

Response: As you recommended, we updated the sentence as follows (Introduction, page 3, lines 57-60):

In the clinical trials of ASOs in HD, Roche's phase III drug Tominersen reduced HTT levels but was halted due to a poor risk-benefit profile, while Wave's drugs (WVE-12010 and WVE-120102) did not induce effective silencing of *mHTT* mRNA.

2. Lines 69-73 feel a bit out of place, perhaps put them after line 80 so it flows better.

Response: Based on your comments, we have revised the points in the manuscript.

3. Line 189: more context needed. "To assess the effect on other CAG repeat regions we looked at *Atxn3* levels.."

Response: As you recommended, we described the additional sentence as follows (Results, page 8, lines 246-249):

In addition, to assess the effect on other CAG repeat regions we looked at *Atxn3* levels, expression of *Atxn3* was significantly lower in Cas9-sgRNA-treated wild-type mice than dCas9-sgRNA- and Cas9 only-treated mice (Supplementary Fig. 6a).

4. Lines 292-301: Altered HTT and mHTT expression in WT fibroblasts, but no effect on WT *Atxn3* levels? The Cas9 is not specific to HTT, so why is dCas9 lowering WT HTT but not

lowering other WT loci? Looking at more loci than just Atxn3 would be advantageous.

Response: As you suggested, we provided the description in the Results as follows:

We propose two explanations for why wtHTT and mHTT expression was altered in WT fibroblasts, but did not affect wild-type ATXN3 levels. This is because 1) dCas9 requires a longer duration compared to Cas9 (which allows genome editing in a short time), and 2) dCas9 will have a relatively more transcriptional repression in the longer CAG repeat region of the mutant allele than the wild-type allele.

In addition, we confirmed several genes containing CAG repeats associated with polyglutamine-associated diseases, such as *ATXN1*, *ATN1*, *ATXN7*, *CACNA1A*, *TBP*, and on-target gene, *HTT* in both human HD and human control fibroblasts. In human control fibroblasts, after Cas9-sgRNA treatment, the expression of *ATXN1*, *ATN1*, and *HTT* genes significantly decreased compared to the Cas9 only- and dCas9-sgRNA treatment. In HD fibroblast containing 40-50 CAG, *ATN1* and *HTT* were significantly suppressed in the Cas9-sgRNA-treated group compared to the other groups and *HTT* gene expression was decreased in the dCas9-sgRNA group than Cas9 only group. The Cas9-sgRNA and dCas9-sgRNA significantly reduced the *HTT* gene in GM21756 cell line (HD fibroblasts) compared to Cas9 only group, and treatment with Cas9-sgRNA suppressed *HTT* gene expression more than treatment with dCas9-sgRNA. In GM09197 fibroblast, expression of *ATN1* and *TBP* genes was significantly suppressed by the Cas9-sgRNA treatment and *HTT* gene expression was significantly decreased in both the Cas9-sgRNA group and dCas9-sgRNA group compared to Cas9 only group. These results showed that suppression of the potential off-target genes was not observed in the dCas9-sgRNA-treated group and suggested that CRISPR-Cas9 cuts the DNA to shut the targeted gene off, but sgRNA targeting CAG repeat regions may have side effects of mutating normal genes. Consequently, both in-silico and qRT-PCR for our sgRNA were used to identify potential off-targets (Supplementary Fig.4) (Results, pages 6, lines 149-166).

5. Line 300: " These results suggest that CRISPRi does not significantly regulate the expression of normal gene with CAG repeats." You do report lowered expression of WT HTT with both systems though.

Response: We agree with your comments. We revised the sentence as follows (Discussion, page 12, lines 366-370):

When we investigated the expression of genes containing CAG repeats such as *ATXN1*, *ATN1*, *ATXN3*, *ATXN7*, *CACNA1A*, and *TBP* in human HD fibroblasts, the expression of the genes was significantly reduced by Cas9-sgRNA treatment, but not by dCas9-sgRNA treatment. These results suggest that CRISPRi unlikely to significantly regulate the expected off-target genes expression with CAG repeats.

REVIEWERS' COMMENTS:

Reviewer #1 (Remarks to the Author):

The authors have addressed my remaining questions.